# Subsiding shells and the distribution of up- and downdrafts in warm cumulus clouds over land

Christian Mallaun[1], Andreas Giez[1], Georg J. Mayr[2], and Mathias W. Rotach[2]

[1]Deutsches Zentrum für Luft- und Raumfahrt (DLR), Flight Experiments, Oberpfaffenhofen, Germany
[2]University of Innsbruck, Institute of Atmospheric and Cryospheric Sciences, Innsbruck, Austria

*Correspondence to:* Ch. Mallaun
(christian.mallaun@dlr.de)

**Abstract.** The mass flux of air lifted within the updrafts of shallow convection is usually thought to be compensated outside the cloud through either large scale subsidence or stronger downdrafts in a thin shell surrounding the cloud. Subsiding shells were postulated based on large eddy simulation and are experimentally tested in this study for shallow convection over land. Isolated cumulus clouds were probed with a small research aircraft over flat land and mountainous terrain, in different wind situations and at different levels of the clouds. The average of the 191 cloud transects shows the subsiding shell as a narrow downdraft region outside the cloud boundaries. The ensemble-mean subsiding shell is narrower on the upwind side of the cloud, while it is at least half a cloud diameter wide and more humid on the downwind side. At least half of the upward mass transport in the cloud is compensated within a distance of $20\%$ of the cloud diameter. However, this shell is not uniform. Distinct regions of downdrafts and updrafts with high variability of the vertical wind are frequent and randomly distributed in the vicinity and also within the cloud. The median diameter of the drafts directly at the cloud boundary is at least 4 times as large as inside the clouds and in the environment. Downdrafts at the cloud boundary are twice as frequent as updrafts. In contrast to the updrafts the major part of the downdrafts is situated outside of the cloud. The subsiding shell results from the distribution of these up- and downdrafts.

## 1 Introduction

Air in shallow cumulus clouds is transported towards higher regions of the atmosphere where it detrains from the cloud and mixes with environmental air. This is an effective way to vertically transport energy, heat and moisture from the surface to higher levels. Traditionally, large scale subsidence between the isolated cloud cells is regarded to be responsible for compensating the mass flux within the cloud (e.g. Stull, 1988). Heus and Jonker (2008) found a characteristic thin layer of a downward airflow outside of the simulated cumulus clouds by means of Large Eddy Simulations (LES), which they named the subsiding shell. This general concept is illustrated in Fig. 1.

A similar concept already appears in the cloud model of shallow cumulus clouds by Scorer and Ludlam (1953). They describe a region of downward motion in the wake of a rising bubble, which is caused by evaporation of the cloudy boundaries. With respect to the turbulence in the cloud they conclude, that the disturbances within the undiluted updrafts might be small compared to the wake region where violent eddies are dominating. Jonas (1990) found such significant downdrafts outside of growing

cumulus clouds from airborne measurements, while these were missing in the decaying clouds. This is also confirmed by later measurements (e.g. Rodts et al., 2003; Blyth et al., 2005).

Wang et al. (2009) investigated the mean dynamical properties of the cloud margin in shallow convection with a large number of cloud transects from aircraft measurements and confirm the subsiding shell as a distinct minimum of vertical velocity at the cloud boundaries. Mixing of cloud and environmental air leads to evaporative cooling, which is the source for the subsiding shell (Heus and Jonker, 2008; Abma et al., 2013; Katzwinkel et al., 2014). Even though the subsiding shell is rather thin, the covered area is significant as it surrounds the entire cloud (Heus and Jonker, 2008). Therefore, the area of the shell is large enough to account for major parts of the downward mass flux in the cloud free environment, while the contribution of subsidence outside of the shell is less important. Jonker et al. (2008) calculated the fraction of mass compensation to be $80\%$ within a diameter of $400\,\mathrm{m}$ around a cumulus cloud. The ability of the clouds to condition the entire atmospheric boundary layer (ABL) is strongly reduced by these downdrafts. Additionally, they are an efficient way to bring air from the top of the cloud to its lateral boundaries, where it can entrain into the cloud. Consequently, this entrained air has properties from above the entrainment level. Wang and Geerts (2010) showed that the thermodynamic properties of the air in the vicinity of the cloud vary strongly with its horizontal distance from the cloud.

Most measurements discussed so far targeted shallow convection above the ocean (e.g. Heus and Jonker, 2008; Jonas, 1990; Katzwinkel et al., 2014), although this cumulus cloud type is also a common and characteristic phenomenon in the temperate and continental climate of the mid-latitudes. The measurements of Katzwinkel et al. (2014) were restricted to the top level of the investigated maritime cumuli because of system limitations. They investigated the individual updrafts at the cloud top and therefore found mostly small clouds. Wang et al. (2009) included shallow convection over land in their analysis and investigated the mean properties of the cloud ensemble. Many of their convective clouds over land were containing rain and ice particles with cloud tops usually well above $4\,\mathrm{km}$. In this study, we present the results of 6 measurement flights over central Europe to test the validity of the subsiding shell to further detail over different types of land characteristics (flat versus mountainous terrain). The cloud transects were flown at different height levels, directions as well as during different synoptic situations. Thus, the analysis is limited to cumulus humilis and mediocris, but allows for a comprehensive picture of these cloud types.

We investigate the dynamical properties of these clouds with a special focus on the cloud borders looking for the subsiding shell. Due to the turbulent character of the cloud system the subsiding shell can only be detected in the mean distribution of the vertical wind near the cloud boundaries. As the individual cloud transects are known to include strong up- and downdrafts within the cloud (Yang et al., 2016), we expand the analysis of these drafts to the cloud boundaries and the near environment, so as to understand the structure of the subsiding shell.

In the following section we describe the assets and limitations of the instrumented aircraft and give an overview of the measurement campaign and methods. In Section 3 we present the results for some selected cloud transects. This is followed by more general observations of the mean properties and variability of shallow cumulus clouds, the characteristics of the subsiding shell and the distribution of up- and downdrafts. We discuss the importance of the subsiding shell with a focus on the downward mass flux and the statistics of the drafts in Sect. 4, before we end with the conclusions in Sect. 5.

## 2 Probing shallow convection and the subsiding shell

### 2.1 The research aircraft

For the in situ measurements we used a Cessna Grand Caravan 208B (Caravan), which is equipped with a meteorological sensor package (Mallaun et al., 2015). This small research aircraft combines several advantages for the investigation of small scale phenomena in the ABL such as the strong single engine power, high manoeuvrability and robust design. It is equipped with a high accuracy inertial reference system (IRS) for position and attitude determination and a meteorological sensor package mounted under the left wing. Mallaun et al. (2015) describe the details of the measurement instrumentation and the corresponding uncertainties for the high-frequency $100\,\mathrm{Hz}$ measurements of pressure, temperature, humidity and wind vector. The main results of the measurement accuracy are summarized in Table 1.

### 2.2 The measurement campaigns

We conducted 6 measurement flights during two campaigns in June 2012 and July 2013 as listed in Table 2. Flights 1, 2 and 6 were conducted over relatively flat terrain north of the Alps and west of Munich (D), with smooth hills covered by fields and woodland. On the first two flight days high pressure influence was dominating. The wind and wind shear were moderate from western direction during flight 1 and very weak during flight 2. Examples for the clouds during the first two flights are shown in Figure 2. During flight 6 the wind was moderate from north-west and in the rather humid surrounding ($rh > 70\,\%$) the cloud cover was higher and the cumulus clouds were situated in lower levels compared to the other flight days. Flights 3 to 5 were devoted to the investigation of convective clouds over alpine orography. The clouds developed above the mountain peaks during strong high pressure influence with weak southerly wind. The convection tended to start above distinct points above the mountain ridges drifting north during its life cycle. The flight tracks are shown in Figure 3 and information about the flight conditions can be found in Table 2.

We chose a similar flight strategy for all flights in order to achieve comparable data sets. Each flight started and ended with a vertical profile to obtain information about the undisturbed atmosphere outside the cloud. During ascent the cloud base and top were defined visually and a mean wind direction was estimated from the on-board quicklook data. With this information the operator defined the flight directions 'along' and 'across' the mean wind and up to three height levels within the cloud. In some cases also transects below cloud level were flown. Figure 4 shows the definitions of flight levels and directions as well as the main flight pattern, which is shaped like an 8. We also performed a simple reverse heading pattern, which allows for a high transect rate and facilitates the relocation of the target cloud. Beside the single cloud sampling we also performed longer straight flight legs in different directions and levels in order to gain broader statistics of the cloud properties.

### 2.3 Identifying clouds

The target clouds were selected visually during the flight. The identification of the cloud boundaries is realized in two steps. First, a digital time mark set by the operator during the flight gives a rough estimate of the location. As a second step, we

take the signal of relative humidity to determine the exact cloud boundaries. Thus, *the cloud starts and ends with humidity saturation* as measured by a Ly-$\alpha$ absorption hygrometer (Buck, 1976), which has a response time faster than the acquisition frequency of $100\,\text{Hz}$.

We request a cloud diameter of at least $200\,\text{m}$ to avoid very small cloud filaments. Such a cloud transect typically includes about 300 data points. This limit left us with 191 cloud transects including 17 different individual clouds which were repeatedly penetrated. Other authors have required different minimum cloud lengths. The scarce resolution of models or earlier measurements required higher thresholds of $\approx 500\,\text{m}$ (e.g., Heus and Jonker, 2008; Jonas, 1990). More recent measurements, for example Wang et al. (2009) request a minimum length of $200\,\text{m}$, or Katzwinkel et al. (2014) one of $50\,\text{m}$.

Several factors complete the identification of a cloud. A single cloud often consists of more than one updraft. It can contain large gaps above its base, which makes it difficult to distinguish it from other clouds in the vicinity. Figure 2 a) shows an example. The cloud consists of an active updraft near the upwind side of the cloud separated by a gap at higher levels from an older, already decaying updraft further downwind, but joined through a common cloud base. For the data evaluation, we have used the flight protocol and video tape to confirm the common cloud base. We also use a subset of 94 transects for which gaps in the transects above common cloud base were at most $150\,\text{m}$ and less than $30\,\%$ of the cloud diameter. The cloud definition is summarized in Table 3. The existence of cloud gaps is in line with recent measurements (e.g. Jonas, 1990; Blyth et al., 2005; Wang et al., 2009; Katzwinkel et al., 2014). The detection of the common cloud base it hardly possible with a fully automatic cloud analysis, but inevitable with our observations during the measurement campaign.

We classified the cloud transects in terms of cloud region (bottom, middle, top), along- or crosswind transects and terrain (lowland, mountains). A further criterion regards the activity status of the cloud, where we request a positive median buoyancy inside the cloud for active clouds. The numbers of selected cloud transects representing the different criteria are listed in Table 4.

No agreement exists what constitutes a subsiding shell. Heus and Jonker (2008) originally defined a $50 - 100\,\text{m}$ range of negative vertical wind directly outside the cloud. Wang et al. (2009) use a range of $50\,\text{m}$ within and $200\,\text{m}$ outside the cloud. Katzwinkel et al. (2014) split the subsiding shell in an inner and outer shell, where the inner shell has negative vertical velocities and negative buoyancy. It is driven by the negative buoyancy after mixing and evaporation at the cloud boundary (Abma et al., 2013) and thus, can partially also appear inside the cloud. The outer shell has still negative vertical velocity but positive buoyancy. Generally, the existence of the subsiding shell is identified by a negative peak of the mean (or median in a non-Gaussian process) vertical velocity right outside the cloud boundaries. Due to the turbulent character of the cloudy environment a single representation of a cloud transect will usually not exhibit the characteristics of a subsiding shell. The mean distribution of the vertical velocity gives insight in the strength and depth of the subsiding shell. We investigate the width of the shell relative to the cloud diameter which accounts for the strong variability of cloud size. A circular subsiding shell with a width of $20\,\%$ in cloud diameter has an area approximately equal to the embedded cloud.

In order to assess the structure of the subsiding shell we analyse the properties of the up- and downdrafts in the vicinity of the cloud. In accordance with Yang et al. (2016) we define a downdraft (updraft) as the region where the vertical velocity is below $-0.2\,\text{ms}^{-1}$ (above $+0.2\,\text{ms}^{-1}$). The small deviation from zero accounts for small-scale turbulence inside the up- and

downdrafts and also corresponds to the measurement uncertainty of the system. Thus, regions with small vertical velocity are disregarded. We omit up- and downdrafts narrower than $10\,\mathrm{m}$. Furthermore, where the gap between two neighboring downdrafts (updrafts) is smaller than $10\,\mathrm{m}$ and the vertical wind does not exceed $+0.2\,\mathrm{ms^{-1}}$ (fall below $-0.2\,\mathrm{ms^{-1}}$) the up- or downdraft is considered as a single one. They are estimated for the cloud region and up to $0.5$ cloud diameters away of the

5 cloud boundary. Three different categories of up- and downdrafts are distinguished: inside the cloud, at the cloud boundary and in the environment. The up- or downdraft at the cloud boundary are situated partly inside *and* outside the cloud.

## 2.4 Computation of derived variables

### 2.4.1 Corrections of measurements in clouds

The presence of liquid water in the cloud modifies temperature and humidity measurements. Some of the liquid water evapo-

10 rates as air is compressed in and in front of the total air temperature housing reducing the static temperature ($T_s$) and increasing the humidity mixing ratio (r) and thus the dewpoint temperature ($T_d$). We can estimate $T_d - T_s$ as the sum of evaporative cooling ($\Delta T_s$) and the increased dewpoint temperature ($\Delta T_d$) with

$$T_d - T_s = \Delta T_s + \Delta T_d = \frac{L_h \cdot \Delta r}{c_p} + \frac{\partial T_d}{\partial r} \cdot \Delta r, \tag{1}$$

as long as no significant sub- or supersaturation is present inside the cloud. The bias in water vapour mixing ratio ($\Delta r$) is

15 equal to the evaporated amount of cloud water. In this approximation we use $L_h = 2.5\,\mathrm{MJkg^{-1}}$ for the standard enthalpy of evaporation and $c_p = 1005\,\mathrm{JK^{-1}kg^{-1}}$ for the heat capacity at constant pressure. The change of dewpoint temperature with the change of mixing ratio ($\partial T_d / \partial r$) depends on pressure and temperature.

The humidity mixing ratio correction can be computed from Eq. 1,

$$\Delta r \approx (T_d - T_s) / \left( 2.5\,\mathrm{K\,g^{-1}\,kg} + \frac{\partial T_d}{\partial r} \right), \tag{2}$$

if the mixing ratio is expressed in $\mathrm{g\,kg^{-1}}$, where $(T_d - T_s)$ is measured and the value for $\frac{\partial T_d}{\partial r}$ is calculated individually for each flight as listed in Table 5 following the common approximations for humidity conversion (e.g. Stull, 2000). The evaporation of $\Delta r$ causes a cooling of the static temperature ($\Delta T_s$) of

$$\Delta T_s = \frac{L_h \cdot \Delta r}{c_p} \approx 2.5\,\mathrm{K\,g^{-1}\,kg} \cdot \Delta r. \tag{3}$$

This correction rarely exceeds $1\,\mathrm{K}$ for the temperature and $0.4\,\mathrm{gkg^{-1}}$ for the mixing ratio.

However, when sensor wetting occurs as described by Lawson and Cooper (1990) and Wang et al. (2009), a cold peak can cause significantly larger errors especially outside the cloud and this correction does not work. On the Caravan two redundant temperature sensors (identical in construction) were available, which show different sensor wetting and thus, also different amplitudes of the cold peak. This allows for a very simple detection of the wetting effect. Consequently, for the investigation of the potential temperature and buoyancy distributions we have used just the first half of the transects in order to minimise

the impact of sensor wetting. As the transects can start on either side of the cloud, the median distributions are available for the entire cloud transects, but contain a reduced set of data. The cold peak was often not visible in our measurements and the corrections defined in Eqs. 2 and 3 are applied to all data.

### 2.4.2 Computation of buoyancy

The buoyancy is determined according to

$$B = g \left[ \frac{\Theta'_v}{\overline{\Theta_v}} + (1 - \kappa) \frac{p'}{\overline{p}} - 10^{-3} r_l \right], \tag{4}$$

(Eq. 2.52, Houze (2014)) . To determine the virtual potential temperature ($\Theta_v$) in clouds, the liquid water content (LWC) is additionally needed (i.e., $\Theta_v = \Theta(1 + 0.61 \cdot 10^{-3} r - 1 \cdot 10^{-3} r_l)$, with the liquid water mixing ratio ($r_l$) (Stull, 2000). Again, $r$ and $r_l$ are expressed in $\mathrm{g\,kg^{-1}}$. Since the LWC is not measured directly, we omit this effect in the calculation, which introduces

a positive bias within the clouds. However, this bias will be small for the shallow cumulus clouds especially at the cloud boundaries where we find the region of our special interest. In order to estimate the bias, we calculated an adiabatic value of LWC as the difference of the saturation humidity mixing ratio at the measurement height and the cloud bottom. We estimated an increase of the LWC to be $dLWC \approx 2\,\mathrm{g\,kg^{-1}km^{-1}}$ for the measurement flights described in Sec. 2.2. Only for flight 6 it was smaller with $dLWC \approx 1.6\,\mathrm{g\,kg^{-1}km^{-1}}$. According to Warner (1977) the true LWC is much smaller and will rarely exceed

$r_l = 1\,\mathrm{g\,kg^{-1}}$. Thus, also the contribution to the buoyancy is small.

Similar to Wang et al. (2009), we calculate the mean values ($\overline{\Theta_v}$) and mean pressure ($\overline{p}$) from the data of each cloud transect. The perturbation values ($\Theta'_v, p'$) are then defined as the deviation from these mean values. In Equation 4, $\kappa$ is the ratio of the gas constant and the specific heat capacity of air at constant pressure (i.e., $\kappa = R/c_p = (c_p - c_v/c_p)$) and $g$ the acceleration due to gravity. The conserved variable $\Theta_v$ is used to compensate for inevitable height changes of the aircraft during the passage

through the cloud. The pressure is altitude-corrected as described in Mallaun et al. (2015) with

$$p_{\mathrm{ref}} = p_0 \cdot e^{-\frac{g \cdot \Delta h}{R \cdot \overline{T_v}}}. \tag{5}$$

For $p_0$ we take the pressure at the starting point and $\overline{T_v}$ is the mean value of virtual temperature approximated by the mean values at the current position and the starting point, $\Delta h$ is measured with the DGPS.

### 2.4.3 Computation of the vertical mass flux

In order to calculate the mean vertical mass flux ($f_m$) from the center of the cloud to the cloud boundary and the compensating downward directed mass flux outside of it, we adopt the formulation presented in Yang et al. (2016). From the flight data only the mass flux along the flight track can be estimated, the differences compared to an areal approach are discussed in Heus et al. (2009). We calculate the vertical mass flux for the distance ($x$) from the cloud boundary with

$$f_m(x) = \rho(x) \cdot w(x) \cdot \mathrm{d}x. \tag{6}$$

$w(x)$ is the vertical velocity at the position $(x)$ and $\rho(x)$ the air density. The accumulated mass flux $(F_m)$

$$F_m(x) = \int\limits_{x_0}^{x} f_m(x')\,\mathrm{d}x' \tag{7}$$

measures the integrated upward flux of air inside the cloud and estimates the compensating downward mass flux outside. The limits of integration range from the cloud center $x_0$ to $x$. In our analysis we consider only relative values of $f_m(x)$ and $F_m(x)$, which are scaled by their respective maximum values. Also the horizontal distance $x$ is scaled to the individual cloud length. Thus, the smaller clouds have the same statistical weight as the big clouds when the averages for all the cloud transects are calculated.

## 3 Properties of the cumulus clouds and the subsiding shells

Altogether, we investigated 191 cloud transects for the measurement flights described in Sec. 2.2. The clouds are selected according to the cloud definition in Table 3 and results in 94 transects when the stricter criteria are applied. An overview on the numbers for the different transect classification is given in Table 4. All these transects build a large sample to investigate the statistical distribution of the characteristic cloud properties. The boundaries of the clouds are estimated by the humidity distribution. Thus, the dynamical properties in the focus of the following discussion are independent of the cloud definition. First, we look at a series of particular cloud transects during flight 2. This helps to explain the methods as well as to discuss the cloud characteristics and the subsiding shell for the chosen examples.

### 3.1 The vertical wind distribution in individual cloud transects

During the day of flight 2 shallow convection formed around midday in a low-wind situation with weak high pressure influence. Compared to the other flight situations the horizontal wind and wind shear of $\approx 1\,\mathrm{ms}^{-1}\mathrm{km}^{-1}$ were very weak - at least up to the highest flight level. Some meteorological parameters of the environmental air are listed in Table 2.

Figure 2 a) shows an example cloud of this flight including a narrow cloud turret, which grew fast above the broader and longer persisting cloud base. After $5 - 10\,\mathrm{min}$ the turret (the upper part of the cloud) dissolved in the relatively dry surrounding air and gave way to a new updraft, while the cloud base persisted.

Figure 5 shows measurements along a crosswind transect flown in the upper part of another cloud during flight 2. The relative humidity in panel a) shows a compact cloud with small cloud gaps in the western part indicated by subsaturation. Here, also the vertical wind velocities (panel b) are small compared to the eastern half where updrafts of up to $5\,\mathrm{ms}^{-1}$ are present. Also the buoyancy shown in panel c) is increased in the updraft region, while the pressure perturbation in panel d) is significantly negative in the dissolving (or decaying) part of the cloud.

Outside the cloud boundaries, a clear signal of sinking air with magnitudes up to $3\,\mathrm{ms}^{-1}$ is present. On the left boundary an $\approx 200\,\mathrm{m}$ wide region of downdrafts starts already within the cloud. On the right side the downdraft region is $\approx 300\,\mathrm{m}$ wide with a distinct minimum about $150\,\mathrm{m}$ away from the cloud boundary followed by a weak subsidence region. It is important to

note, that due to the turbulent character of the cloudy environment the representation of a single cloud transect cannot give a distinct information about the existence of the subsiding shell.

Not many of the investigated individual transects possess such distinct downdrafts directly outside of both of the cloud boundaries. For example, Fig. 6 shows humidity and vertical wind for 4 different transects for the same cloud in north-south direction
(along the main wind direction). From the video tape and operator's notes there is strong evidence that all cloud parts have a common base, even though rather large sub-saturated regions occur (e.g., Fig. 6 c). Such gaps occur very frequently when weaker decaying cloud parts and regions with stronger updrafts tend to line up along the mean wind direction. It is almost impossible to recognise the vertical wind structure from one transect to the other, which is due to the turbulent nature and the high spatial /temporal variability and transient behaviour of the flow in the cloud. Apparently, not even the updraft (downdraft)
regions can be identified as quasi-steady 'coherent structures' as is sometimes the case in small-scale turbulent flows. However, in panel c) and d) the main updraft might be the same, but for the rest of the transects the vertical velocity structures are different. This is similar for many transects in other clouds (not shown).

Figures 5 and 6 exemplify the large variations of strength and diameter or distance of the downdrafts in the vicinity of the cloud boundaries. We also find updrafts or vast regions of downdrafts near the cloud. Downdrafts are also frequent within the cloud
itself, especially in the vicinity of cloud gaps (see Fig. 6 c near position 0.25).

We find a similar distribution of the vertical wind also for the transects of the other flights. The turbulent character of the cloud environment is obvious. The up- and downdrafts seem to be rather randomly distributed with strong up- and downdrafts within the cloud as well as in the environment.

### 3.2 Distributions of humidity, wind, pressure and buoyancy

Figure 7 shows the median vertical velocity distribution for all the cloud transects. Note that the spatial coherence of the individual transects is lost with the representation of the percentiles. The median vertical velocity has a distinct maximum within the cloud, which is slightly shifted towards the upwind side. The vertical wind minimum outside of the cloud boundary is the subsiding shell. The vertical velocity becomes already negative well within the cloud. Thus, the average cloud boundary experiences downward motion. The minimum slightly outside of the cloud boundaries is stronger on the downwind side. Further
away from the cloud the downdrafts become weaker. The 75 and 90 percentiles have no downdrafts at all while the 10 and 25 percentiles show continuous negative vertical velocity. The minimum near the cloud boundary is visible for all percentiles, but is weaker for the 75 and 90 percentiles.

Figure 8 shows the median vertical wind distribution for different cloud categories stratified by cloud activity, level within cloud, underlying terrain and along or crosswind transects. The $95\%$ confidence interval was computed at each point along
the scaled transect by bootstrapping (1000 repetitions with replacement) and is shown in gray. Even taking the uncertainty resulting from the limited sample size into account, the median vertical velocities for all active transects in Fig. 8 a), except the bottom one, are clearly distinguishable between the interior and exterior of the cloud. Such a distinction is not possible for the inactive transects shown in Fig. 8 b), especially since even bootstrapping will underestimate the uncertainty due the smaller sample sizes for this class (Efron, 1979).

Active clouds (except at the bottom level) have pronounced updraft regions and a subsiding shell at the boundaries. The strongest updrafts are found at cloud top level. The most distinct downdraft regions at the cloud boundaries are present on the downwind side of the transects of the center level and the clouds above mountains. They have a broad region of sinking air, which already starts well within the cloud. Looking at the active crosswind transects we find this wind minimum as well,

but here the vertical wind almost vanishes within half a cloud diameter. The upwind side of the active along wind transects have almost no downdrafts with a very narrow minimum right outside the cloud boundary. The inactive transects show high variability of the wind signals inside and outside of the cloud. At cloud mid and top level they do not show any strong updrafts. Figure 9 shows the histograms of the vertical velocity inside the cloud and within $20\%$ outside of the cloud diameter. The distributions obviously differ in size and shape, with some statistical values summarized in Table 6. In the cloud the mean

vertical velocity is $\approx 0.5\,\mathrm{m/s}$ and the skewness of the distribution is directly visible in the figure with increased frequencies of fast rising parcels. However, only one of the selected transects has no negative vertical velocity at all within the cloud. Except of 8 cases all the transects have downdrafts stronger than $-1\,\mathrm{m/s}$ inside the cloud. In the shell the mean vertical velocity is significantly below zero for all four cloud boundaries. Especially on the upwind side the distribution is narrow compared to the other investigated parts. In the downwind and crosswind shells we find stronger downdrafts and higher variability compared

to the upwind side. The highest variability of the vertical velocity is present within the clouds, which is also visible in Fig. 7. In Fig. 9 a) the stronger downdrafts in the downwind shell compared to the upwind shell become visible. The frequencies and magnitude of the updrafts are similar for the shell region on both sides. A separated analysis of the left and right crosswind shells does not lead to any significant differences neither for the median distributions nor for the histograms.

Figure 10 presents the median distribution of the relative humidity and horizontal alongwind perturbation as well as the buoy-

ancy and the horizontal pressure perturbation for the 191 selected cloud transects. The median relative humidity within the cloud is saturated, but the 10 percentile is significantly below. Due to the definition in Table 3 all values are $rh = 100\%$ at the cloud boundaries. Outside of the cloud boundaries the relative humidity decreases rapidly. The gradients are stronger on the upwind side and the median value is significantly enhanced on the downwind side for at least half a cloud diameter, which can be explained by the humidity halo on the downwind side of the cloud (Perry and Hobbs, 1996). The mean horizontal wind

component along the flight track ($u_{ac}$) in Fig. 10 b) is significantly reduced within the cloud, where also the strongest updrafts are found. For the 10 and 25 percentiles this signal is most pronounced. It is enhanced on the upwind side and matches the mean values on the downwind side. This feature is only present in the alongwind transects, while it is not visible in the crosswind transects. It is strongest in the bottom level transects and vanishes in the top level (not shown). The distribution of $u_{ac}$ is also characterized by a high variability which is similar to the vertical wind variability.

Figure 10 c) shows that within the cloud a mean upward motion coincides with enhanced buoyancy, while on both sides outside of the cloud the buoyancy is almost zero on average. On the upwind side a weak negative peak is indicated with a strong and clear gradient through the cloud boundary. This gradient is much weaker on the downwind side of the cloud, where values near zero are present well within the cloud. The median pressure perturbation (Fig. 10 d) is small and with magnitudes of a few pascals similar to the sensor resolution ($= 2\,\mathrm{Pa}$). A weak negative anomaly is visible within the cloud, which is counteracted

by positive contribution especially on the upwind side of the cloud. However, the percentiles show that significant deviations of the hydrostatic equilibrium are frequent both inside and outside of the clouds.

### 3.3 Sensitivity of the results

Even though the clouds were actively chosen during the flight with a focus on vital clouds, many of them contain big cloud gaps. Different rising plumes, decaying cloud parts with strong downdrafts and also subsaturated air parcels entrained into the cloud coexist and build the entity of a single cloud. From the chosen cloud transects 9 cases have no cloud gaps at all. For 25 cases the fraction of cloud gaps relative to the cloud diameter exceeds $50\%$. For the 25 percentile, the median and the 75 percentile we estimate a cloud fraction of $\approx 10\%$, $\approx 20\%$ and $\approx 40\%$, respectively.

In order to judge the robustness of the results in terms of cloud definition, we have repeated the analyses for the stricter criteria including restriction 4 and 5 as defined in Table 3. Thus, we omit the transects with a fraction of cloud gaps of more than $30\%$ or a cloud gap exceeding $150\,\mathrm{m}$. For the new analysis we select the more homogeneous clouds and neglect the less active or more complex ones, that just 94 out of 191 cloud transects remain. In Table 4 the numbers of total occurrences are listed by the numbers in brackets. In Figure 7 the respective median distribution for the reduced sample of 94 'ideal' clouds is represented by the green line. It is obvious, that neglecting the less active clouds leads to stronger updrafts. However, the distribution at the cloud boundary and in the cloud free region remains almost unchanged. Also the histograms of the vertical velocity (not shown) remain qualitatively unchanged. The frequencies of the vertical velocities within the cloud are shifted towards higher values. The vertical velocity distribution in the shell regions are narrower compared to the results in Fig. 9. After all, the selection of the clouds does not substantially change the results.

### 3.4 Distribution of updrafts and downdrafts

In Figure 7 we find a pattern similar to a subsiding shell on both sides of the cloud boundaries. However, due to the turbulent character of the cloudy environment the subsiding shell is not visible in most of the individual transects (as shown e.g. in Figs. 5 and 6). Instead, we find a large variety of up- and downdrafts of different strength and size inside the cloud, in the environment and also around the cloud boundaries. According to the definition in Sec. 2.3 we analysed the 191 cloud transects and found 1735 downdrafts and 1495 updrafts. This corresponds to an average of 9 different downdrafts and 8 updrafts for each cloud transect including half a cloud diameter around the clouds. Within the clouds, the average number of up- and downdrafts is approximately equal ($\approx 4$ of each), but the updrafts are about $25\%$ larger. In the environment, on the other hand, downdrafts are slightly more frequent and larger than the updrafts. Directly at the cloud boundaries, finally, the downdrafts are twice as frequent and larger as compared to the updrafts. There are only 328 up- and downdrafts in the cloud boundary region and in 56 cases no significant up- or downdraft was identified. The increased frequency of the downdrafts at the cloud boundary leads to the subsiding shell in the mean distribution of the vertical wind. A summary of the exact numbers and the main properties is given in Table 7. Most remarkably, the median sizes of the drafts are much larger directly at the cloud boundary compared to the other regions. There, also the median and variance of the vertical wind are strongest. This indicates that we find a large number of relatively small drafts within the cloud and also in the environment. These smaller drafts are less frequent at the

cloud boundary which leads to the characteristic distribution shown in Fig. 11. The distributions of the drafts within the cloud and in the environment are very similar, only the frequency of large diameters is reduced in the environment. The distribution within the cloud corresponds well to the results of Yang et al. (2016). They find some larger drafts (i.e., $D > 2\,\mathrm{km}$) than in our sample, because our investigation is limited to shallow convection. The distribution at the cloud boundary is different. While

at the cloud boundary the small diameters are rare, sizes of several hundred meters are most frequent for the downdrafts and somewhat larger for the updrafts. The largest of these updrafts often cover significant portions of the cloud and can form the main (i.e., the largest) updraft of the cloud. In Table 7 also the statistics for the main updrafts as a subset of the cloud updrafts are listed. These main updrafts have a median length similar to the drafts at the cloud boundary and a strength and variability of the vertical wind that is higher compared to the other categories.

The numbers of downdrafts at the cloud boundary are almost equally distributed around the cloud but they have smaller diameters at the upwind side compared to the crosswind and downwind sides as listed in Table 8. The updrafts are slightly more frequent on the upwind side. They are smallest at the crosswind side and twice as large on the downwind side. While the major parts of the downdrafts lie outside of the cloud, the updrafts are situated more inside. Table A1 in the appendix provides detailed information about the mean properties of the up- and downdrafts at the cloud border with respect to the different

transect categories. A comparison of the draft diameter and median vertical velocity in the scatterplot of Fig. 12 a) shows that for all three categories larger drafts often have stronger vertical winds. For the larger diameters above $D > 200\,\mathrm{m}$ the smaller magnitudes of the vertical velocity are most often in the environment. The distribution and the high variability of the larger drafts inside the cloud and at the boundary are very similar, but the occurrences of smaller drafts at the cloud boundary is clearly reduced. In the comparison of the median vertical velocity and buoyancy in Fig. 12 b) it becomes clear that updrafts

and downdrafts have both, positive and negative buoyancy and thus the median values are small. The drafts in the cloud have more cases with positive and the drafts in the environment more with negative buoyancy. However, it is clearly visible that the negative buoyancy is more frequent with downdrafts and positive buoyancy is more frequent with updrafts. The exact values are given in Table 7. There, also the correlation coefficients of the vertical velocity and buoyancy are listed. The correlation for the downdrafts is small and at the cloud boundary even slightly negative. Thus, stronger downdrafts do not necessarily have

more negative buoyancy. The updrafts have a higher correlation except for the cloudfree environment.

## 4    Discussion

The median vertical velocity distribution presented in Fig. 7 agrees well with results of former analyses of the subsiding shell (e.g. Heus and Jonker, 2008; Wang et al., 2009; Katzwinkel et al., 2014). Different to earlier works, in the current study we considered only shallow convection over land, captured transects in all cloud levels and included also rather complex clouds (i.e,

the clouds can have several updrafts and cloud holes, as long they have a common cloud base). The vertical velocity possesses a distinct minimum directly outside of the cloud boundaries, which is associated with a shell of sinking air covering the entire cloud. Figure 13 shows the relative vertical mass flux ($f_m$) and the relative accumulated mass flux ($F_m$) from the cloud center outwards. The vertical mass flux is calculated with Eq. 6, which leads to a very similar distribution and magnitude as the vertical

velocity. Mathematically, the vertical wind signals are weighted with the horizontal resolution and the air density, which in the most cases lies near $1\,\mathrm{kg\,m^{-3}}$. The maximum of mass flux is found well within the cloud, while a distinct minimum exists right outside of the cloud boundary. The downward flux near the cloud boundary has almost the same strength as the upward flow in the main updraft region. Half of the downward mass flux along the transect occurs within a distance of $20\,\%$ of the cloud diameter outside of the cloud. After half a cloud diameter the mass flux in the cloud is compensated. Both distributions of $f_m$ and $F_m$ are very similar to the observations of Heus et al. (2009), even though the clouds over land often have complex structures and include cloud gaps. Different to their results the vertical mass flux becomes negative already well within the cloud where already a significant portion of downward mass flux occurs. This is obvious with the vertical wind distribution that becomes negative inside the cloud boundaries as well. There is no significant change in the results, when we restrict the analysis from all the 191 cases to the 130 alongwind transects as shown by the grey dashed lines in Fig. 13 or to the crosswind transects (not shown).

So far, our results corroborate the findings of Heus and Jonker (2008). However, care must be taken when interpreting the mean distributions of cloud and shell properties. While a significant downdraft anomaly - the subsiding shell - is present in the median vertical wind distribution (see Fig. 7), this is not a characteristic feature of each individual cloud. There is a strong variability of the vertical wind outside of the clouds and the position of the downdrafts (and also the updrafts). Although downdrafts are frequent near the cloud boundaries and also within the cloud itself, they often do not form a coherent shell around the cloud surface. Instead, these downdrafts alternate with updrafts of similar strength and diameter. The consecutive legs in Fig. 6 show how fast the wind structures change around the evolving cloud. These turbulent eddies are responsible for the vertical mass transport as well as for the entrainment of environmental air into the cloud. The presence of a subsiding shell is the result of averaging the highly variable up- and dominating downdrafts near the evolving cloud. Thus, the composition of the drafts directly at the cloud boundary form the subsiding shell. In order to understand the origin of the subsiding shell we have to look at the distribution of the up- and downdrafts. At the cloud boundary the downdrafts are twice as frequent compared to the updrafts which leads to the characteristic distribution of the vertical wind. The downdrafts have a larger diameter at the downwind and crosswind sides, which explains the weaker signal of the subsiding shell on the upwind side. Table 8 shows that significant portions of the up- and downdrafts are situated each in and outside of the clouds, which indicates the connection of the air masses to both sides of the cloud boundary. Compared to the other regions these drafts have much larger diameters, which shows the importance of the (turbulent) exchange processes at the cloud boundary. Within these up- and downdrafts, where cloudy air as well as environmental air is present, several processes are important. Most obvious is the influence due to mixing of air parcels and evaporation, but also the drag of adjacent air masses, the pressure gradient force or radiation can play a role (Park et al., 2017). Heus et al. (2009) found the evaporative cooling responsible for the subsiding shell. An indication for the evaporation at the cloud boundary is the enhanced humidity visible directly outside of the cloud in Fig. 10 a) which results very probably from evaporating cloud droplets. Same as for the downdraft velocities this effect is stronger on the downwind side of the cloud compared to the upwind side. Wang et al. (2009) and Katzwinkel et al. (2014) find the negative buoyancy near the cloud boundary as an indication for the droplet evaporation which drives the sinking shell. The results in Fig. 12 and Table 7 show the same relation also for the up- and downdrafts. While most of the updrafts have positive buoyancy, it is negative for

about 70 % of the downdrafts. However, the stronger downdraft does usually not indicate a lower buoyancy.

Downdrafts are frequent also inside the clouds and have a significant influence on the mass flux. About one third of the upward directed mass flux is already compensated inside the cloud by the downdrafts. Half of these in-cloud downdrafts have a negative buoyancy and one third has significant subsaturation ($rh < 95\,\%$).

As a main conclusion from the analysed cloud transects over land, we find the dominating downdrafts directly at the cloud boundary to be the origin of the subsiding shell. These drafts have a median diameter of $\sim 20\,\%$ in cloud diameter (see Table 7). Defining this as the subsiding shell, its area is approximately equal to the embedded cloud. This 'subsiding shell' is a valid concept for *ensembles* of clouds as shown in Fig. 7. According to Fig. 8 the subsiding shell is typical for active clouds, most pronounced in the center and top cloud regions or for the crosswind transects. The subsiding shell is more pronounced for the

transects over the mountains compared to the flat land. A comparison with the results given in Table A1 shows, that for these categories the downdrafts are not more frequent and also not much stronger, but they have a larger diameter. Additionally, there is a reduced number of updrafts for the mountain, middle layer and crosswind transects compared to the transects over the land, the cloud top and alongwind. The former have much smaller diameters and weaker updrafts. Thus, the subsiding shell is not only defined by the intensity of the downdrafts, but also by the distribution and development of the updrafts at the cloud border.

In Figure 8 the difference of the vertical wind distribution between the active and inactive cloud transects is striking. For the inactive transects the updraft region as well as the subsiding shell are missing. The differences of the up- and downdrafts at the cloud border of the inactive transects are less pronounced compared to the other categories. The frequency, the strength of the vertical wind and also the portion outside of the cloud (i.e., dry part) are similar. However, the variance of the vertical wind and size of the updrafts are smaller.

Our results show (see Fig. 13), that the mass transport in the cloud is compensated within half a cloud diameter away from the cloud boundary. This has strong implications for the distribution and mixing of the cloud air in the environment. Compared to the concept of a downward mass flux via subsidence (Stull, 1988), less mixing and less transport of heat and energy occur. The mixing of cloud air in the upper ABL is reduced when the air stays near the cloud and directly sinks down in the subsiding shell to lower regions. Thus, the 'subsiding shell' has to be considered in a parametrisation scheme for shallow convection over

land.

## 5   Conclusions

A series of cloud transects measured with a research aircraft were analysed with a special focus on the dynamical properties near the cloud boundaries. Former LES model results had shown a narrow coating downdraft region around shallow convective clouds, which is called a subsiding shell.

To test whether the subsiding shell can be observed for shallow convection over land, we conducted 6 measurement flights in the years 2012 and 2013. It was possible to probe single clouds over flat land and mountain ridges, in different heights and different synoptic situations. The aircraft measured the thermodynamic properties of the clouds with the exception of liquid water content. A correction is presented for the temperature and humidity bias that occurs due to droplet evaporation inside

the clouds. The target clouds were actively selected during the flights in order to choose well-defined vital clouds. For the investigation we manually selected 191 cloud transects. The clouds are usually not homogeneous masses of cloud air with a central main updraft but more complex formations with regions of updrafts, downdrafts and cloud gaps within one cloud. With a stricter cloud definition we repeated the analysis with a reduced cloud sample of 94 'ideal' clouds for a sensitivity test.

The median vertical velocity of the selected cloud transects shows a very similar distribution compared to the LES model results. We also do not see any significant differences between our measurements over land surface compared to earlier results from shallow convection over sea. The main feature in the distribution is a distinct minimum in the vertical wind immediately outside of the cloud boundaries. A distinct downdraft on the downwind side starts well within the cloud and is wider compared to the upwind side, where the gradients of vertical velocity and buoyancy are stronger. A strong downward mass flux is present

in the region of the subsiding shell, which compensates for a large fraction of the positive vertical mass transport within the cloud. Within a distance outside the cloud of $\approx 20\,\%$ in cloud diameter half of the upward directed vertical mass flux is compensated.

In general, the distribution of the vertical wind is qualitatively similar over flat land and mountainous terrain, but there are quantitative differences. Active clouds have larger vertical velocity and vertical mass flux than inactive clouds. The strongest

updrafts are present in the upper level and crosswind transects, while the downdrafts are most pronounced at the center level and mountain transects.

Due to the turbulence in the environment of the clouds, the subsiding shell is not visible in the individual cloud transects. Strong downdrafts are twice as frequent in the vicinity of the cloud boundaries compared to the updrafts, which leads to the characteristic feature of a subsiding shell in the mean vertical velocity profile. The individual cloud transects are characterized by strong

updrafts and downdrafts both, inside and outside the cloud. They seem rather randomly distributed which is expectable for turbulent eddies with sizes much smaller than the cloud diameter. Compared to the cloud and the environment the diameters of the up- and downdrafts at the cloud boundary are much larger with stronger and more variable vertical wind. Only the strongest updrafts inside of each cloud lead to a similar distribution of size and strength. The drafts at the cloud boundary have a median diameter of $\sim 20\,\%$ in cloud diameter and form the subsiding shell. In the middle layer, above the mountains and crosswind

the downdrafts have the largest diameters, they are strongest at the cloud tops and above the mountains. The striking difference of the vertical wind distribution between the active and inactive cloud transects (i.e., the inactive clouds do not show a distinct updraft region nor the subsiding shell) is not directly visible for the up- and downdrafts at the cloud border. The majority of the downdrafts at the cloud boundary have negative buoyancy and the relative humidity is increased compared to the cloudfree environment, which both indicates the importance of evaporative cooling for the formation of the subsiding shell. However,

the reason for the dominating sizes of the drafts directly at the cloud boundary cannot yet be explained and remains an open question for future research. Finally, the concept of the subsiding shell seems a valid concept for mean properties of shallow convection over land with all its implications on the cloud air mixing and entrainment of upper level air into the cloud. The downdraft in the subsiding shell is able to account for a major part of the downward mass flux, which is compensating the net upward mass flux in the cloud. In contrast to subsidence in a large area between the clouds this process reduces the horizontal

mixing of cloud air in the upper boundary layer, but keeps the cloud air in the near vicinity of the cloud itself.

*Data availability.* Data on personal request

*Competing interests.* All authors declare that they have no competing interests.

*Acknowledgements.* We thank the DLR flight crew for their enthusiastic commitment while circling in narrow turns around the target clouds. We thank the air traffic authorities in Germany and Austria for their support, which gave the permission to freely sample the clouds in these regions of high air traffic. Furthermore, we thank the two anonymous reviewers for their insightful comments which largely improved the paper.

## Appendix: Characteristics of up- and downdrafts at the cloud border

Table A1 is an expansion of Table 7. It contains the detailed information of the up- and downdrafts at the cloud border separately for the different transect categories.

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

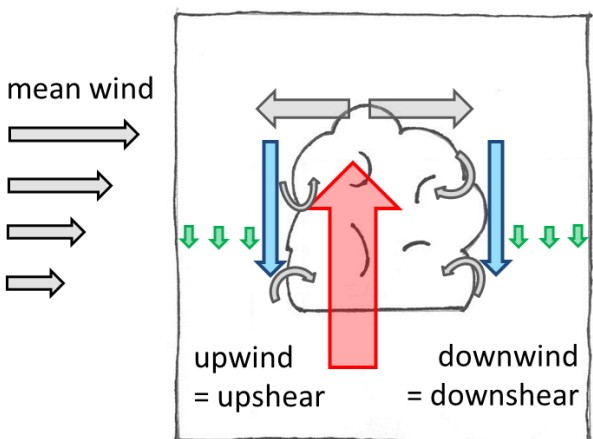

**Figure 1.** Conceptual model of a small cumulus cloud. The vertical mass flux within the cloud (red arrow) is compensated either through large scale subsidence (green arrows) or in the subsiding shell (blue arrows). Grey arrows indicate detrainment above the cloud and entrainment on the lateral cloud boundaries. The main updraft is shifted towards the upshear cloud boundary.

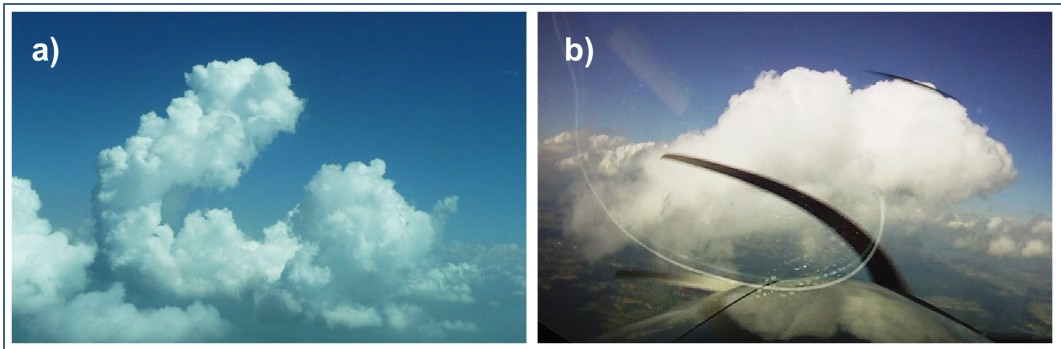

**Figure 2.** Examples of clouds in weak and strong shear environments, respectively. (a) Cloud in weak-wind, weak-shear environment during flight 2. It has a common cloud base, but cloud gaps in the upper part, which is surrounded by drier air $rh \approx 60\%$. Weak winds blow in the lower cloud part with $\approx 2-4\,\mathrm{ms}^{-1}$, the inclination of the cloud top indicates increasing wind with height. (b) Cloud in a boundary layer with strong wind shear during flight 1 immediately before a crosswind transect. The wind blows from the left and the shear-induced declination of the cloud is visible. The cloud bottom does not show a sharp line, which indicates that the cloud has reached at least a mature state, without a strong updraft in the lower cloud parts.

**Table 1.** List of the measurement uncertainties for the main meteorological parameters of the sensors flown on the Caravan research aircraft. Results from Mallaun and Giez (2013)

| Quantity | Variable | $\sigma$ |
|---|---|---|
| Static air temperature | $ts$ | $0.15\,\mathrm{K}$ ($0.5\,\mathrm{K}$ in clouds) |
| Humidity mixing ratio | $mr$ | $2\%$ ($4\%$ below $0.5\,\mathrm{g/kg}$) |
| Relative humidity | $rh$ | $3\%\mathrm{rh}$ ($5\%\mathrm{rh}$ below $0.5\,\mathrm{g/kg}$) |
| Dewpoint temperature | $T_d$ | $0.35\,\mathrm{K}$ ($0.5\,\mathrm{K}$ in clouds) |
| Angle of attack | $\alpha$ | $0.25°$ |
| Angle of sideslip | $\beta$ | $0.25°$ |
| Wind speed | $ws$ | $0.3\,\mathrm{m/s}$ |
| Wind angle | $wa$ | $2°$ |
| Alongwind component | $u_f$ | $0.3\,\mathrm{m/s}$ |
| Crosswind component | $v_f$ | $0.3\,\mathrm{m/s}$ |
| Vertical wind | $w$ | $0.25\,\mathrm{m/s}$ |

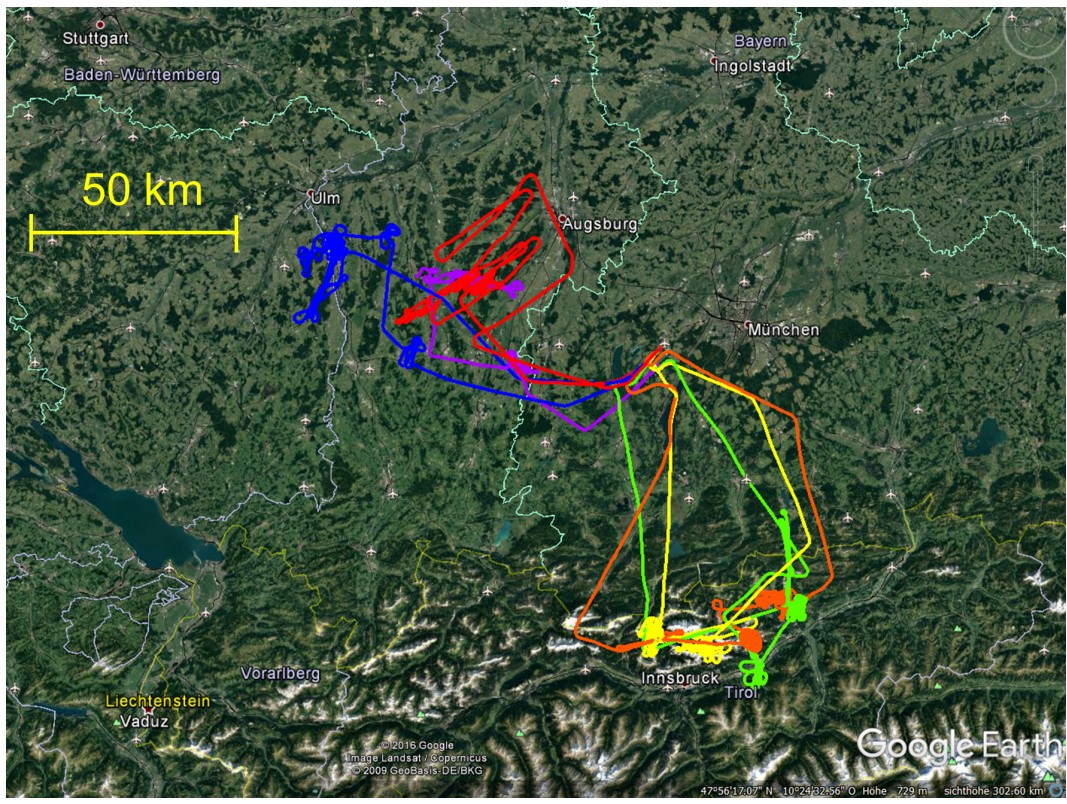

**Figure 3.** Overview of the target region for the measurement flights above the northern Limestone Alps and foothills west of Munich. The lines show the 6 flights listed in Table 2 colored blue, red, orange, yellow, green and purple, respectively. The thin yellow line marks the border between Austria and Germany ( 2016 Google, Image Landsat / Copernicus).

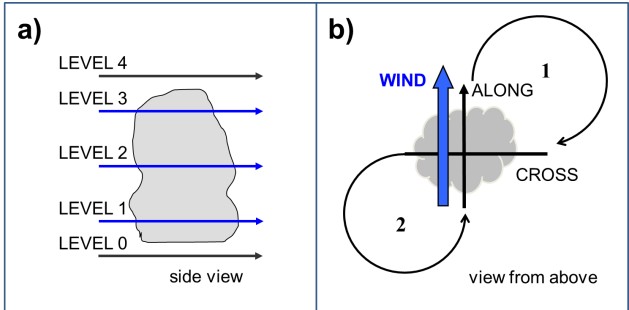

**Figure 4.** Definition of the chosen levels (a) and directions (b) during the measurement flights. The turns 1 and 2 in panel (b) indicate the main flight pattern resembling the number '8', which results in repeated flight transects along and cross the mean wind.

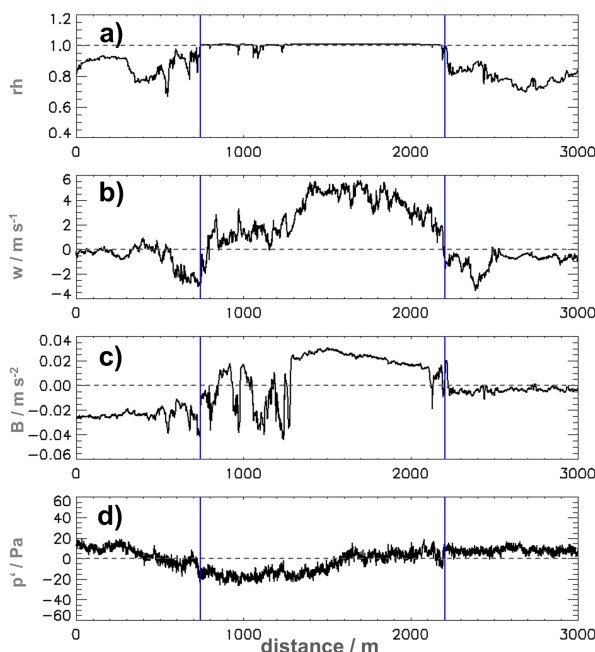

**Figure 5.** Measurement values for a crosswind transect through an active cloud during flight 2 looking downwind. The cloud boundaries are marked by the blue vertical lines. Panel (a) shows relative humidity; (b) vertical wind; (c) buoyancy without the contribution of LWC and (d) the horizontal pressure perturbation.

**Table 2.** Summary of flights conducted during the measurement campaigns in June 2012 and July 2013, with the number of cloud transects used in this study (191 total) and their pressure height measured in hecto feet. The given values for the environmental air correspond to the lowest and highest flight level, respectively. The lifted condensation level (LCL) is estimated with the Henning equation (e.g., Schmeissner et al., 2015) from the profile data measured during takeoff and landing at the airport about $80\,\mathrm{km}$ north of the target area.

| number | date | time [UTC] | number of transects | flight levels [hft] | temperature [°C] | wind [ms$^{-1}$] | relative humidity | LCL [m] |
|---|---|---|---|---|---|---|---|---|
| 1 | 10/07/2012 | $12:30-14:58$ | 38 | $75, 80, 90$ | $7-3$ | $10-16$ | $85-80$ | 2050 |
| 2 | 26/07/2012 | $8:15-10:45$ | 47 | $70, 80, 90$ | $11-5$ | $2-4,$ | $70-55$ | 2100 |
| 3 | 18/06/2013 | $11:20-14:10$ | 30 | $115, 120, 130$ | $4-0$ | $4-6$ | $65-40$ | 3050 |
| 4 | 19/06/2013 | $11:29-14:15$ | 35 | $120, 130$ | $3-0$ | $8-10$ | $60-50$ | 3250 |
| 5 | 20/06/2013 | $11:26-14:08$ | 22 | $120, 130$ | $3-0$ | $6-6$ | $60-50$ | 2600 |
| 6 | 26/06/2013 | $9:00-11:36$ | 19 | $60$ | $-1$ | $7$ | $75$ | 1400 |

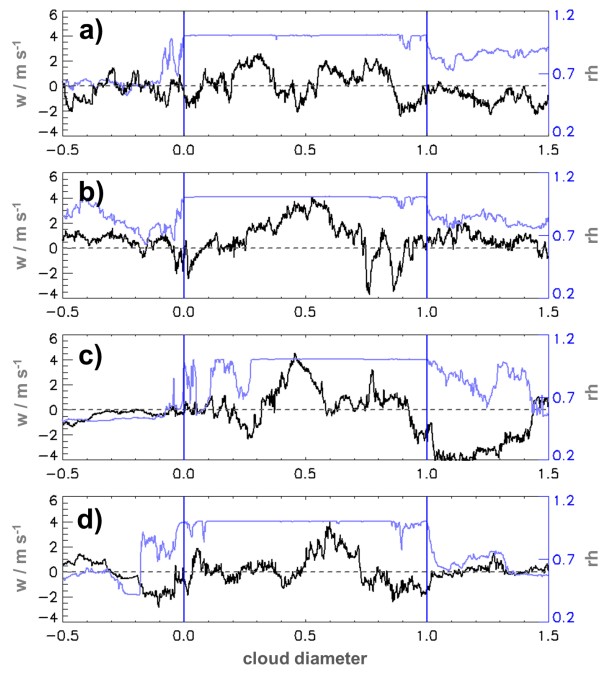

**Figure 6.** Relative humidity (blue line) and vertical wind (black line) for four alongwind directed cloud transects of an individual cloud during flight 2. The blue vertical lines indicate the cloud boundaries. The x-axis is scaled to the horizontal diameter of the cloud, where 0 marks the cloud edge on the upwind side and 1 the downwind edge. The data outside the cloud are shown for half a cloud diameter, each. The starttime of the transect, cloud length and height of the flight level are for panel (a) 12.27 UTC, 1043 m and 2620 ma.s.l., panel (b) 12.33 UTC, 1561 m and 2620 ma.s.l., panel (c) 12.40 UTC, 772 m and 2920 ma.s.l., panel (d) 12.42 UTC , 673 m and 2940 ma.s.l..

**Table 3.** Criteria for identifying the cloud. The stricter cloud requirements 4 and 5 are optional and used in a repetition of the analysis in order to test the sensitivity of the results.

|  | Cloud criteria: |
| --- | --- |
| 1. | The cloud boundaries are defined by reaching humidity saturation. |
| 2. | A cloud has a minimum diameter of 200 m. |
| 3. | All parts of a single cloud possess a common cloud base, thus, a cloud transect can also contain regions of subsaturation (cloud gaps) |
| (4. | Any region of subsaturation (cloud gap) is shorter than 150 m.) |
| (5. | The cloud gaps may not cover more than 30 % of the cloud diameter) |

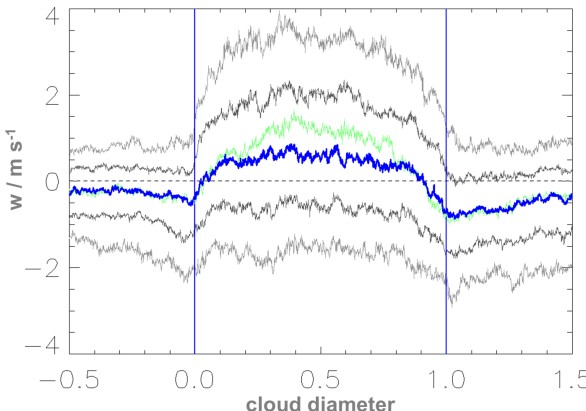

**Figure 7.** Distribution of the vertical wind speed of 191 cloud transects: median (blue line) $10, 25, 75$ and $90$ percentiles (grey lines) with the scaling of the x-axis and the cloud boundaries as in Fig. 6. The individual cloud transects are scaled by the cloud length. The transects are arranged in a way that the upwind side is on the left and the crosswind transects are shown from left to right. The vertical blue lines indicate the cloud boundaries. The green solid line is the median of the vertical wind velocity for 94 selected cloud transects, which fulfill the stricter cloud requirements in Table 3.

**Table 4.** Characteristics of the 191 (94) selected cloud transects as defined in Table 3. Numbers in parentheses are relative to the subset of 94 clouds with stricter limits on the cloud gaps. The transects are divided into legs along and cross to the main wind direction, into legs at the bottom, center or top of the cloud and the activity status. Active clouds have a positive mean buoyancy inside the cloud.

|  | along | cross | flat land | | mountain | |
|---|---|---|---|---|---|---|
|  |  |  | active | inactive | active | inactive |
| total | 130 (60) | 61 (34) | 85 (49) | 19 (6) | 68 (37) | 19 (2) |

|  | bottom | | center | | top | |
|---|---|---|---|---|---|---|
|  | active | inactive | active | inactive | active | inactive |
| total | 10 (8) | 3 (2) | 80 (47) | 14 (5) | 63 (31) | 21 (1) |

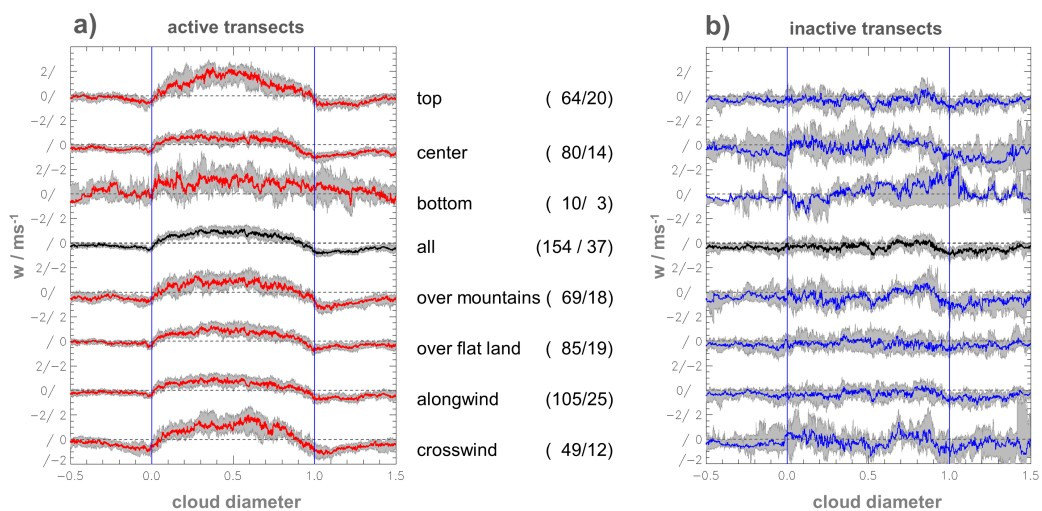

**Figure 8.** a) Median of the vertical wind for different transect heights, terrain, direction for the active cloud transects. The comparison is based on the 191 cloud transects shown in Fig. 7, with the same scaling of the x-axis. The red lines show active clouds. The detailed selection is explained in between the two panels of the respective line including the number of involved cases (i.e., active / inactive cases). For better readability the lines are vertically shifted and the grey dashed horizontal lines show the different 0 lines. Two adjacent 0 lines are separated by $4\,\mathrm{ms}^{-1}$. In order to show the statistical significance of the transect samples a statistical resampling (bootstrapping method) with 1000 repetitions is performed. The resulting spread of the $95\,\%$ confidence interval for each is transect category is shown by the grey shaded area. b) same for the inactive transects (blue lines).

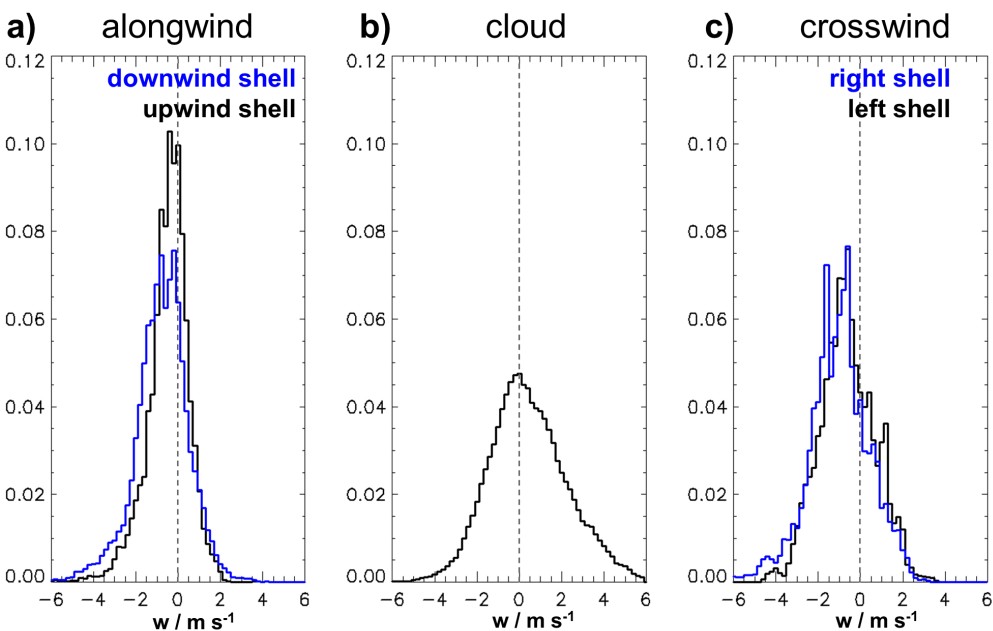

**Figure 9.** Distribution of the vertical wind in the cloud and shell regions for the 191 cloud transects. The three panels show the probability density function for a) the upwind shell and the downwind shell; b) the cloud and c) the right shell /left shell for the crosswind transects. For the distribution we set a bin size of $0.2\,\mathrm{m\,s^{-1}}$ and the results are scaled with the number of data points. The width of each shell is set to $20\,\%$ of the respective cloud diameter.

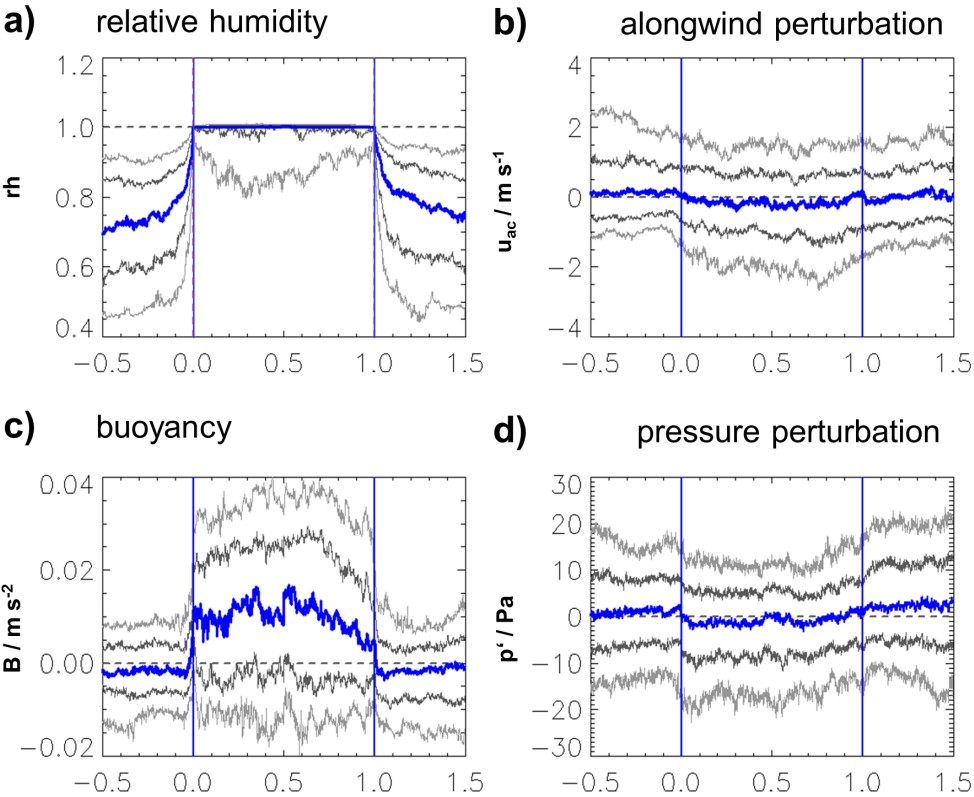

**Figure 10.** Same as Fig. 7 for the relative humidity (a), horizontal wind perturbation of the along flight path component (b), buoyancy (c) and the horizontal pressure perturbation (d).

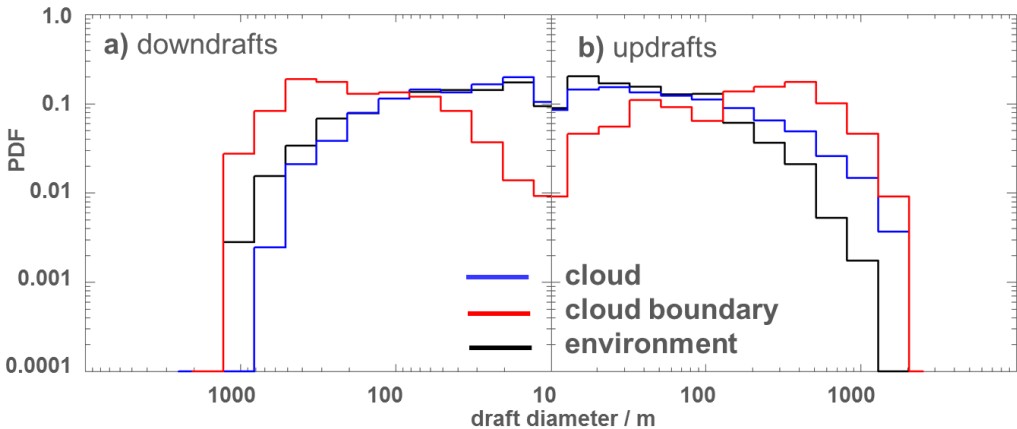

**Figure 11.** PDFs of the diameters of the updrafts and downdrafts. The distributions are shown separately for the drafts inside of the cloud, at the cloud boundaries and the near environment.

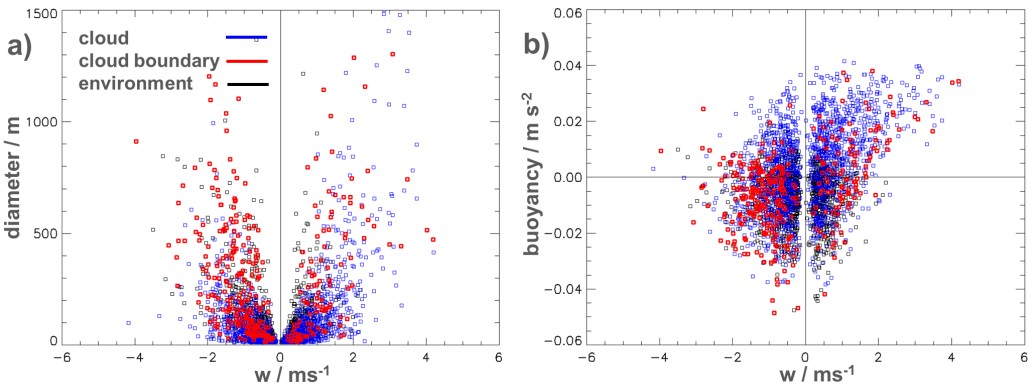

**Figure 12.** a) Scatterplot of median vertical velocities ($w$) and diameters for the up- and downdrafts in the cloud, at the cloud boundary and the environment. Each point represents one individual draft. b) Same for the median vertical velocities ($w$) and buoyancy.

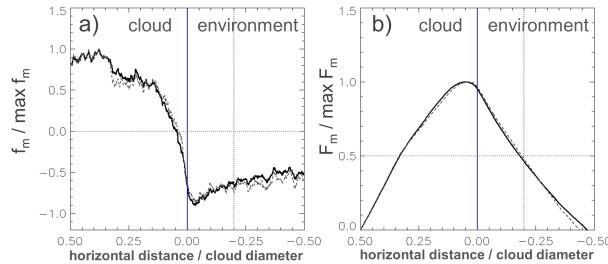

**Figure 13.** (a) Mean vertical mass flux ($f_m$) along 191 cloud transects scaled with the maximum mass flux. The x-axis is scaled with the cloud diameter. The grey dashed line shows the scaled $f_m$ for 130 alongwind transects. (b) Integrated mass flux ($F_m$) from the center of the cloud (i.e., Eq. 7) scaled with the maximum value. The dotted vertical line in both panels indicates the position of $20\,\%$ in cloud diameter where $\approx 50\,\%$ of the upward massflux is compensated by the subsiding shell.

**Table 5.** Change of saturation dewpoint temperature in dependence of water vapor mixing ratio for different dewpoint temperatures ($TS$) and pressures ($PS$) during the measurement flights. The last column gives the estimated average value which is used for the temperature correction described in Sect. 2.4

| flight | $FL$ | $PS$ | $TS$ | $\frac{\partial T_d}{\partial r}$ | $\overline{\frac{\partial T_d}{\partial r}}$ |
|---|---|---|---|---|---|
| | [hft] | [hPa] | [°C] | [K g$^{-1}$kg] | |
| | 75 | 770 | 7 | 1.8 | |
| 1 | 85 | 735 | 5 | 1.9 | 2.0 |
| | 95 | 705 | 2 | 2.2 | |
| | 75 | 770 | 10 | 1.5 | |
| 2 | 85 | 735 | 8 | 1.6 | 1.6 |
| | 100 | 700 | 5 | 1.8 | |
| | 125 | 630 | 5 | 1.6 | |
| 3-5 | 130 | 625 | 4 | 1.7 | 1.8 |
| | 140 | 595 | 1 | 2.0 | |
| 6 | 65 | 800 | 2 | 2.5 | 2.5 |

**Table 6.** Vertical wind speeds [ms$^{-1}$] of 191 selected cloud transects. The length of the cloud interior is variable and the shells are limited to $20\%$ of the cloud diameter. The table shows the mean vertical velocities, the median as well as the 25 and 75 percentiles.

|  | mean | 25 percentile | median | 75 percentile |
|---|---|---|---|---|
| cloud | +0.5 | −0.7 | +0.4 | +1.7 |
| upwind shell | −0.4 | −0.9 | −0.3 | +0.2 |
| downwind shell | −0.7 | −1.4 | −0.6 | +0.1 |
| crosswind shell | −0.8 | −1.6 | −0.7 | +0.2 |

**Table 7.** Numbers and median properties of the downdrafts and updrafts selected from the 191 cloud transects according to the definition in Sec. 2.3. The drafts inside the cloud, at the cloud boundary and in the environment are investigated separately. The median value is listed for the absolute length and relative to the cloud diameter (rel. length), the vertical velocities and the variance of the vertical velocity. This is followed by the fraction of positive buoyancy ($B > 0\,\mathrm{ms}^{-2}$) and finally the correlation coefficient ($r_{wB}$) of buoyancy and vertical wind.

|  |  | numbers | length [m] | rel. length [%] | vertical wind [ms$^{-1}$] | variance [m$^2$s$^{-2}$] | $B > 0\,\mathrm{ms}^{-2}$ [%] | $r_{wB}$ |
|---|---|---|---|---|---|---|---|---|
| downdrafts | all | 1735 | 58 | 4 | −0.7 | 0.10 | 34 | 0.03 |
|  | cloud | 810 | 46 | 3 | −0.7 | 0.11 | 42 | 0.12 |
|  | cloud border | 217 | 223 | 19 | −1.1 | 0.28 | 29 | −0.22 |
|  | environment | 708 | 52 | 4 | −0.5 | 0.05 | 25 | −0.02 |
| updrafts | all | 1495 | 57 | 4 | 0.6 | 0.09 | 52 | 0.48 |
|  | cloud | 813 | 58 | 4 | 0.7 | 0.16 | 71 | 0.44 |
|  | cloud border | 109 | 233 | 17 | 1.1 | 0.32 | 55 | 0.54 |
|  | environment | 573 | 45 | 3 | 0.4 | 0.04 | 24 | 0.08 |
|  | cloud main updraft | 179 | 290 | 24 | 1.5 | 0.5 | 88 | 0.55 |

**Table 8.** Number, median length and median relative portion of the drafts outside of the cloud (dry part) for the drafts at the cloud border. Subsets are shown for the drafts on the upwind, downwind and crosswind sides of the cloud.

| | numbers | length [m] | dry part [%] |
|---|---|---|---|
| downdrafts | | | |
| upwind | 69 | 155 | 75 |
| downwind | 76 | 240 | 57 |
| crosswind | 72 | 287 | 73 |
| updrafts | | | |
| upwind | 43 | 201 | 31 |
| downwind | 35 | 347 | 35 |
| crosswind | 31 | 159 | 42 |

**Table A1.** Similar Table 7 but for the different transect categories at the cloud border. The last two columns are missing, instead the portion of the dry part as in Table 8 is listed.

|  |  | numbers transects | numbers drafts | length [m] | rel. length [%] | vertical wind [ms$^{-1}$] | variance | dry part [%] |
|---|---|---|---|---|---|---|---|---|
| downdrafts | cloud border all | 191 | 217 | 223 | 19 | −1.1 | 0.28 | 69 |
|  | top | 64 | 71 | 191 | 18 | −1.3 | 0.35 | 79 |
|  | middle | 80 | 91 | 254 | 20 | −1.1 | 0.25 | 62 |
|  | bottom | 10 | 8 | 100 | 8 | −0.7 | 0.18 | 78 |
|  | land | 85 | 98 | 184 | 17 | −1.0 | 0.25 | 66 |
|  | mountain | 69 | 72 | 341 | 31 | −1.2 | 0.37 | 75 |
|  | along | 105 | 112 | 187 | 17 | −1.1 | 0.27 | 67 |
|  | cross | 49 | 58 | 287 | 27 | −1.2 | 0.35 | 77 |
|  | all inactive | 37 | 47 | 273 | 19 | −1.1 | 0.24 | 61 |
| updrafts | cloud border all | 191 | 109 | 233 | 17 | 1.1 | 0.32 | 35 |
|  | top | 64 | 38 | 181 | 22 | 1.2 | 0.50 | 39 |
|  | middle | 80 | 40 | 270 | 22 | 1.2 | 0.31 | 24 |
|  | bottom | 10 | 11 | 360 | 23 | 0.8 | 0.22 | 62 |
|  | land | 85 | 54 | 290 | 24 | 1.2 | 0.33 | 29 |
|  | mountain | 69 | 35 | 172 | 15 | 1.0 | 0.33 | 42 |
|  | along | 105 | 662 | 270 | 18 | 1.2 | 0.33 | 36 |
|  | cross | 49 | 23 | 233 | 18 | 1.1 | 0.41 | 42 |
|  | all inactive | 37 | 20 | 95 | 8 | 1.1 | 0.23 | 35 |