# Peer review of "Subsiding shells and the distribution of up- and downdrafts in warm cumulus clouds over land"

_Atmospheric Chemistry and Physics, 2018_

## Referee Comment (RC1) · Anonymous Referee #1 · 14 Nov 2018

acp-2018-825 SUBSIDING SHELLS AND VERTICAL MASS FLUX IN WARM CUMU-LUS CLOUDS OVER LAND by Christian Mallaun, Andreas Giez, Georg J. Mayr, and Mathias W. Rotach

Recommendation: Reject.

This observational study examines the dynamical properties at the boundaries of shallow cumulus clouds over land. In particular, it provides evidences of the existence of a thin shell just outside of the cloud that is characterized by subsiding vertical motion and downward mass flux. The results are consistent with other published observational and modeling works, and the topic fits the scope of ACP. However, in general, this manuscript lacks novelty and originality and does not provide new contributions to the studies of cumulus clouds. My specific comments are listed below.

When the authors explained the motivation of this study, they have claimed that they "investigate the mean distribution as well as individual cloud transects" (L18-19, P2). But looking at individual transects are not meaningful because of the turbulent nature of the environment. That's why composite analysis of cumulus transects have been conducted in previous works: Wang et al. (2009) over both trade wind and continental Cu, and Katzwinkel et al. (2014) over trade wind ones. Both of those observational studies have performed a more detailed analysis and provided evidence of a subsiding shell at cloud edge based on a large sample of Cu clouds. In my opinion, Section 3 of this manuscript is more like a case study that explains how the clouds and their boundaries are defined.

In addition, the definition of cloud is not consistent in the text. It's mentioned in the text that the criterion of 100% relative humidity has been used to identify the edges of clouds (Section 2.3). But the cases shown in Figures 5 and 6 are clearly associated with unsaturated air close to at least one identified cloud edge. Apparently, this criterion has not been objectively applied to every sampled cloud and this would significantly impact the results. Take the case shown in Figure 6c as an example. RH does not reach 100% until the x-axis is larger than 0.3. Therefore, I doubt the part of transect (x = [0-0.3]) should be considered as in-cloud region as the authors have done. And the inclusion of clouds incorrectly identified like this would have changed the mean distribution of vertical velocity, buoyancy, and mass flux across the cloud edges.

The major portion of the manuscript is based on statistical analysis over 191 identified cloud transects. However, throughout the text, I did not find any texts that have discussed the statistical uncertainty of the shown results. The mean distributions are important. But are they statistically important based on the sample size? In particular, the sample size of the inactive and bottom cases is only 3 in Figure 8, which creates a significant uncertainty when they are compared with other cases.

I also have some other comments that I think are important. (1) The authors mentioned that the observed downdraft in the subsiding shell compensates the upward mass flux

within the cloud. This conclusion cannot be drawn based on what's shown. The authors could have tested the validation of the statement by investigating if the updraft in cloud is correlated with the sinking motion around cloud. (2) The authors should give specific panel numbers to each panel in Figures 6 and 8, and more importantly, refer to figure numbers when discussing relevant findings. I have found the text hard to follow in many places. Examples include but not limited to these paragraphs: L18-27, P6; L17-27, P8. (3) How are the cloud samples stratified to active and inactive subgroups? It's not clear in the manuscript. (4) It's better to change the right axis (RH) to blue colors for easy reading in Figure 6. (5) Use consistent units.

---

## Author Comment (AC1) · 28 Dec 2018

We would, first of all, like to thank the reviewers for their thoughtful and detailed comments to our manuscript. The major concern of both Referees is about the novelty of our results. We cite some of their key comments in the following and discuss them in detail below.

Referee #1: "The results are consistent with other published observational and modeling works, and the topic fits the scope of ACP. However, in general, this manuscript lacks novelty and originality and does not provide new contributions to the studies of cumulus clouds."

Referee #2 "After reading the manuscript, I have the impression that quite similar anal-

ysis already published in the cited papers has been applied to just another set of cloud data without really new aspects or findings. Therefore, this manuscript does not give new insight in cloud dynamics and unfortunately lacks of any novelty....."

Both referees thus claim the lack of novelty. This is, strictly speaking, indeed true if only one data set is considered to be 'original' (and thus publishable) for anyone topic and thus a confirmation of earlier findings is not considered to be publishable. If 'novelty', however, includes analysis methods and research focus, it doesn't apply for the present paper. The overall goal of the present paper was to confirm (or otherwise) the existence of a subsiding shell – which were first postulated based on Large-Eddy Simulation (LES) results - in real clouds. Furthermore, the study was aimed at investigating the cloud-to-cloud variability of the subsiding shell characteristics and the associated mass transfer. The first aspect has indeed previously been done. To our knowledge the existence of the subsiding shell has experimentally been investigated by Heus et al. (2009), Wang et al. (2009), Katzwinkel et al. (2014). And indeed, Wang et al. (2009) include also shallow convection over land in their analysis. We feel that a confirmation of those results over land would be a valuable additional contribution to the field. In our study, we investigated target clouds under different conditions throughout their life-cycle and carefully selected the cloud transects. Therefore, our results first confirm earlier results and second add further information about the characteristics of the different transect classes.

If we move to the methodological aspects, it seems that our paper is indeed the first to investigate the relation between individual clouds (i.e., transects through individual clouds) and a statistical ensemble of clouds with respect to the characteristics of the subsiding shell. Heus et al. (2008) define the subsiding shell of an average cloud as a shell of subsiding air that is frequently observed around cumulus clouds. They investigate the origin of this shell, which is associated with an area of negative buoyancy (with a lateral size of 50–100 m), while the pressure gradient is again positive at cloud edge. Finally, they suggest a three-layer model where the subsiding shell surrounds the cloud

and separates it from the environment. Katzwinkel et al. (2013) divide the subsiding shell, again of an average cloud, into a turbulent and humid inner shell adjacent to the cloud interior and a non-buoyant, non-turbulent outer shell. All these considerations suggest the existence of a subsiding shell in terms of a thin layer enveloping the cloud. These studies have investigated the statistical properties of clouds – as it corresponds to the output of a LES. The question whether an instantaneous 'snapshot' of an individual cloud - what we obtain when flying a transect through an individual cloud - exhibits those characteristics as well – as suggested in some of the 'sketch plots' (e.g., our Fig. 1) – or to what degree and for which development stage, is at least valuable and addressing it offers new insight in the 'turbulent character' of the subsiding shell. Therefore, in our paper we analyze the number of occurrences of the mean characteristics of a subsiding shell in many individual clouds. We find that in most of the individual clouds (about 85 %) with different development stage, these are not present on both sides of the cloud. Thus, the characteristics of a subsiding shell cannot be discerned from the individual clouds (over land), at least not from a single (instantaneous) cross-section. However, we can confirm the subsiding shell as a statistical (average) property of the cloud, since - similar to earlier results - we find the signature of the subsiding shell in the median distribution of the vertical velocity. This statistical property is the result of frequent downdrafts in the near surrounding of the clouds. These findings are new and considered important. We conclude that in order to explain the subsiding shell (the statistical property) it is necessary to understand the distribution and origin of the individual downdrafts in the vicinity of the cloud. Additionally, from analyzing the vertical mass transfer in the vicinity of the cloud we can estimate the thickness of the subsiding shell (some 20% of the cloud diameter) for what no clear definition had been available previously.

Additionally, Referee 2 raises the question whether the distinction between maritime cloud dynamics and that over land should be made in principle.

Referee #2 "Why should maritime cloud dynamics should significant differ from continental clouds?"

Sensible and latent heat fluxes are crucial for the formation of convective clouds and they are definitely different over land and sea surfaces. Similarly, the depth and diurnal variability of the boundary layer is different. This is particularly true for the boundary layer structure over mountainous terrain (e.g., Serafin et al. 2018, Lehner and Rotach 2018, Rotach et al. 2017) and hence the cloud regimes we investigate among others. Differential heating of the surface is responsible for triggering convective clouds over land (e.g, Cotton, 1992), which is less important over the sea. Investigating the characteristics of individual and average cloud properties over different surfaces thus reveals the impact of the different factors mentioned above on the structure of convective clouds. Moreover, Stevens and Feingold (2009) have suggested that the complex processes in shallow convection need to be investigated for the different cloud regimes so that they eventually might be combined to a common picture. Therefore, an extension of the investigation on the subsiding shell on shallow convection over land – and different land characteristics – seems to be a logical consequence. Even if we do not find important differences between the overall characteristics of convective clouds over land and those reported in the literature over the ocean – not over 'flat land' and neither over mountainous terrain – we think that it is important to conclude that apparently boundary layer characteristics, despite their overall relevance for cloud formation, do not have a significant impact on the average and individual cloud characteristics.

Finally, referee 2 raises the question

Referee #2 "How can you come up with robust conclusions based on only six flights?"

The measurement flights in shallow convection were conducted during two 3-week-long measurement campaigns in two consecutive years. This resulted in 6 successful measurement flights including 191 cloud transects – and we think it is rather the number of investigated clouds and to a lesser degree the number of flights (as long as they cover a substantial range of conditions) that determine the robustness of the results.

With 'order 100' the number of realized transects is clearly not the 'large number' (order ten thousand, say) one usually demands for statistical significance. Still, practical restrictions (flight endurance, etc.) and the comparably large effort for each sample set certain limits. Also, it is noted that our sample size is comparable to that from other investigations in this context (e.g., Katzwinkel et al., 2014, Perry and Hobbs, 1996). It is clear that certain sub-samples in our study, such as 'transects for inactive clouds over mountainous terrain' (see Tab. 4) are much too small - so that we refrain from drawing any conclusions.

We realize that we have not sufficiently emphasized all these aspects in the manuscript and hope that our response could clarify our intention. Furthermore, we are convinced that the manuscript presents novel and important findings about the nature and characteristics of the subsiding shell. The referees will hopefully follow our argumentation and agree to the preparation of a revised version, where we will certainly attempt to improve the manuscript as discussed above. We will be more than happy to also address the remaining (relatively) minor comments by the referees in the revised manuscript.

References

Cotton, W. and Anthes, R.: Storm and Cloud Dynamics, vol. Volume 44, Academic Press, 1st edition edn., 1992.

Heus, T. and Jonker, H. J. J.: Subsiding Shells around Shallow Cumulus Clouds, J. Atmos. Sci., 65, 1003–1018, http://dx.doi.org/10.1175/2007JAS2322.1, 2008.

Heus, T., J. Pols, C. F., J. Jonker, H. J., A. Van den Akker, H. E., and H. Lenschow, D.: Observational validation of the compensating mass flux through the shell around cumulus clouds, Quarterly Journal of the Royal Meteorological Society, 135, 101–112, http://dx.doi.org/10.1002/qj.358, 2009.

Katzwinkel, J., Siebert, H., Heus, T., and Shaw, R. A.: Measurements of Turbulent Mixing and Subsiding Shells in Trade Wind Cumuli, J. Atmos. Sci., 71, 2810–2822,

http://dx.doi.org/10.1175/JAS-D-13-0222.1, 2014.

Lehner, M. and Rotach, M. W.: Current Challenges in Understanding and Predicting Transport and Exchange in the Atmosphere over Mountainous Terrain, Atmosphere, 9, https://doi.org/10.3390/atmos9070276, http://www.mdpi.com/2073-4433/9/7/276, 2018.

Perry, K. D. and Hobbs, P. V.: Influences of Isolated Cumulus Clouds on the Humidity of Their Surroundings, J. Atmos. Sci., 53, 159–174, http://dx.doi.org/10.1175/1520-0469(1996)053<0159:IOICCO>2.0.CO;2, 1996.

Rotach, M. W., Stiperski, I., Fuhrer, O., Goger, B., Gohm, A., Obleitner, F., Rau, G., Sfyri, E., and Vergeiner, J.: Investigating Exchange Processes over Complex Topography: The Innsbruck Box (i-Box), Bulletin of the American Meteorological Society, 98, 787–805, https://doi.org/10.1175/BAMS-D-15-00246.1, https://doi.org/10.1175/BAMS-D-15-00246.1, 2017.

Serafin, S., Adler, B., Cuxart, J., De Wekker, S. F. J., Gohm, A., Grisogono, B., Kalthoff, N., Kirshbaum, D. J., Rotach, M. W., Schmidli, J., Stiperski, I., Vecenaj, Z., and Zardi, D.: Exchange Processes in the Atmospheric Boundary Layer Over Mountainous Terrain, Atmosphere, 9, https://doi.org/10.3390/atmos9030102, http://www.mdpi.com/2073-4433/9/3/102, 2018.

Stevens, B. and Feingold, G.: Untangling aerosol effects on clouds and precipitation in a buffered system, Nature, 461, 607–613, http: //dx.doi.org/10.1038/nature08281, 2009.

Wang, Y., Geerts, B., and French, J.: Dynamics of the Cumulus Cloud Margin: An Observational Study, J. Atmos. Sci., 66, 3660–3677, http://dx.doi.org/10.1175/2009JAS3129.1, 2009

---

## Author Response (AR1)

**Reply to the Comments of Reviewer #1**

We are grateful for your comments. We have already commented on the major issues in the first response "acp-2018-825-response.pdf" from 28th December 2018 where we also explain the strategy for a major revision of the publication.

In the following we will respond to the specific comments (reprint in italic font) of the review.

"When the authors explained the motivation of this study, they have claimed that they 'investigate the mean distribution as well as individual cloud transects' (L18-19, P2). But looking at individual transects are not meaningful because of the turbulent nature of the environment."

We are well aware of the turbulent nature of the environment. We agree that the term 'individual cloud transects' was misleading and omit this discussion in the major revision. The subsiding shell is now defined by the median distribution of the vertical wind (p4 I28-33)

"That's why composite analysis of cumulus transects have been conducted in previous works: Wang et al. (2009) over both trade wind and continental Cu, and Katzwinkel et al. (2014) over trade wind ones. Both of those observational studies have performed a more detailed analysis and provided evidence of a subsiding shell at cloud edge based on a large sample of Cu clouds. In my opinion, Section 3 of this manuscript is more like a case study that explains how the clouds and their boundaries are defined."

Different to earlier works, in the current study we considered only shallow convection over land, captured transects in all cloud levels and included also rather complex clouds (i.e., the clouds can have several updrafts and cloud holes, as long they have a common cloud base). We already discussed the size of our data set in the common response.

A consistent definition of the cloud borders and the methods is given in Sec. 2, while Sec. 3 is indeed a case study in the first place. In the revised paper, we have shortened this chapter. According to the suggestions of Reviewer #2 we have added a thorough analysis of the up- and downdrafts (see Sec. 3.4), which adds a more detailed analysis to the revised paper.

"In addition, the definition of cloud is not consistent in the text. It's mentioned in the text that the criterion of 100% relative humidity has been used to identify the edges of clouds (Section 2.3). But the cases shown in Figures 5 and 6 are clearly associated with unsaturated air close to at least one identified cloud edge. Apparently, this criterion has not been objectively applied to every sampled cloud and this would significantly impact the results. Take the case shown in Figure 6c as an example. RH does not reach 100% until the x-axis is larger than 0.3. Therefore, I doubt the part of transect (x = [0-0.3]) should be considered as in-cloud region as the authors have done. And the inclusion of clouds incorrectly identified like this would have changed the mean distribution of vertical velocity, buoyancy, and mass flux across the cloud edges"

The definition of the cloud borders and an objective method to select them is taken very seriously in the manuscript. Section 2.3 and especially Table 3 give the applied criteria. Point 3 defines one cloud by the common cloud base, which we verified by the video tape and the operator notes in our analysis. In the concrete example of Fig. 6 c) we agree, that at the first

glance the cloud might not start before 0.3 in relative cloud diameter. However, the video tape shows that the narrow cloud fractions indeed belong to the same cloud. Thus, the objective cloud criteria lead to the necessary inclusion of these cloud parts.

We also see the difficulty, that in some cases the cloud gaps are rather big und the structures of the clouds complex. Thus, the analysis is repeated with stricter cloud criteria including only transects with small cloud gaps. The results are discussed in Sect. 4.2 without leading to any significant changes.

"The major portion of the manuscript is based on statistical analysis over 191 identified cloud transects. However, throughout the text, I did not find any texts that have discussed the statistical uncertainty of the shown results. The mean distributions are important. But are they statistically important based on the sample size? In particular, the sample size of the inactive and bottom cases is only 3 in Figure 8, which creates a significant uncertainty when they are compared with other cases."

In the manuscript we discuss the median distribution and as well their variability by means of the distribution of the 10,25,75,90 percentiles. We agree, that an analysis of the significance should be added to the results where appropriate. We also see the difficulty in interpreting the small sample sizes of the inactive and bottom cases in Figure 8. The reason is very simple; there are not more samples in these classes. They are shown in the figure for reasons of completeness, but we do not draw any further conclusions accepting the lack of significance. We tested the significance of the samples via the bootstrapping method for the revised paper (see p.8 I 29-34) and added this analysis to Fig. 8.

(1) "The authors mentioned that the observed downdraft in the subsiding shell compensates the upward mass flux within the cloud. This conclusion cannot be drawn based on what's shown. The authors could have tested the validation of the statement by investigating if the updraft in cloud is correlated with the sinking motion around cloud."

We follow the method of determining the vertical mass flux presented and discussed in earlier publications (e.g., Yang et al. 2016). We see that the method is not explained clear enough and have improved this in the revised version. (p.6 I 25-20 and p.7 I 1-8)

The calculation of the mass flux along a flight path instead of an aerial calculation leads to modified results (Heus et al. 2009). However, it is an appropriate method to understand the air movements in the surrounding of the cloud. In our analysis, we do not set a major focus on the physical numbers of the estimated fluxes, but on the comparability with earlier results and the main consequences on the cloud system. Furthermore, the accumulated mass fluxes are estimated for each individual transect before averaging and thus give indeed information about the correlation of the cloud massflux and the environment.

(2) "The authors should give specific panel numbers to each panel in Figures 6 and 8, and more importantly, refer to figure numbers when discussing relevant findings. I have found the text hard to follow in many places. Examples include but not limited to these paragraphs: L18-27, P6; L17-27, P8."

We have major interest to improve the readability of the text and figures and therefore have improved the figures and citations in the revised manuscript. We also have improved the mentioned paragraphs.

(3) "How are the cloud samples stratified to active and inactive subgroups? It's not clear in the manuscript."

The definition is written on page 4 line 3ff and again in the caption of Table 4. In Chapter 2.3 where we define our methods it says: "A further criterion regards the activity status of the cloud, where we request positive mean buoyancy inside the cloud for active clouds."

**(4)" It's better to change the right axis (RH) to blue colors for easy reading in Figure 6."**

We have changed the colour of the right axis according to the colour of the relative humidity.

**(5) "Use consistent units."**

Even though we concentrated also on the consistency of the units, it is possible that we sporadically missed the optimum choice. We have, however, assessed the manuscript in this respect and hope to have encountered all inconsistent units.

**Reply to the comments of Reviewer #2**

We first of all, thank the reviewer for their thoughtful and detailed comments. We have already commented on the major issues of the review in the first response "acp-2018-825-response.pdf" from 28th December 2018 where we also explain the strategy for a major revision of the publication.

In the following we will respond to the specific comments (reprint in italic font) of the review.

"page 1, line 19: would not call this process "simple"; better "this general concept is illustrated in Fig 1.."

We changed accordingly.

p2, I 6: not sure if one can conclude – based on the cited observations - that the subsiding shell does surround the entire cloud. To my opinion, such conclusion can only be drawn from LES

In Heus et al., 2008 (the first cited reference here) the authors define the subsiding shell as a thin coating shell surrounding the cloud, based on LES. However, we agree that in the given context of our measurements this argumentation might be misleading. Therefore, we thank the reviewer to make this point and have adjusted our wording accordingly (i.e., describing the subsiding shell in terms of the mean distribution of the vertical wind as suggested also by Reviewer #1 and discussed in the first response "acp-2018-825-response.pdf"). (i.e., p. 4 I 28 -30)

p2, I 15 to 20: it is not convincing to me that clouds over land should differ from clouds over the ocean with respect to sub-siding shells. I think one should better motivate why the presented observations are novel and one could get new insight in cloud dynamics

We already addressed this point in the first response "acp-2018-825-response.pdf". The argumentation is changed accordingly in the revised paper (p. 2 I 15-29)

Sec 2.1.& 2.3 One of my main concerns about the observations themselves is the lack of any cloud droplet sensor for a cloud experiment.

We see the advantage of a direct measurement of the liquid water content (LWC) and cloud droplet sensors. It is needed for a calculation of the buoyancy inside the clouds and would also be useful for the investigation of the mixture of cloud and environmental air. We hope to be able to expand the measurement system with such instruments for future research campaigns. The lack of LWC or cloud droplet distribution limited our analysis to the dynamical aspects of shallow convection as discussed in our manuscript.

In terms of the definition of the cloud borders we do not think that a measurement of LWC or cloud droplet numbers is superior to the method presented here. For both alternative measurements we find arbitrary thresholds above zero in literature in order to define the cloud border. This necessarily leads to biases in the cloud border estimation. Furthermore, the reaction time of the sensor is important as well. The Ly-alpha absorption hygrometer used in our analysis measures with 100 Hz and is much faster than most alternatives.

"Another more technical question is if there is a special inlet to avoid droplets entering the Lyman-alpha system, which might influence the readings when leaving the cloud that might bias the data interpretation. I do not generally question the rH measurement but this should be clarified and/or discussed in detail because it is important. I think for the Lyman-alpha there are better and more fundamental references such as Buck et al."

A modified total air temperature housing is used for the air inlet into the humidity channel. From the inlet a tube is leading the air to the sensors, where also temperature and pressure are measured. The inlet is constructed in a way that first the air of the inlet boundary is separated and second the droplets are separated from the flow through the tube. It is known and discussed in Sect. 2.4.1 of the manuscript (p5 I 9-13) that contamination of the sensors can still occur, which has to be considered. This effect will always lead to an increased humidity mixing ratio and decreased temperature and thus, to increased relative humidity where cloud droplets are present at the sensors. Thus, the influence of cloud droplets inside the cloud will not generate subsaturation. Outside the cloud the influence vanishes very quickly when no sensor wetting occurs.

We have changed the citation according to the suggestion. (p4 I 2)

*"It is well known that quite often cumulus clouds are surrounded by almost saturated air (humidity halos) which cannot be distinguished from droplet-free air with your criterion of cloud edges."*

Outside the cloud we estimate a measurement uncertainty of the relative humidity of 3% (see Table 1). The regions of almost saturated air (rh > 95 %) are very limited also in the humidity halos. Even within the cloud, relative humidity often goes rapidly down to significant values of subsaturation (e.g., Figures 5 and 6). On the exit side we usually see a clear gradient of the relative humidity even though in some cases the temperature signal might be influenced by evaporation. Thus, the relative humidity is a very good choice to estimate the cloud border.

In the revised paper we now show the median distribution of the relative humidity around the clouds (Fig. 10 panel a). There, indeed the humidity halos are visible, but also the very narrow regions of almost saturated air.

Sec 2.2: At some point it would be essential to get more information about the sampled cloud fields, e.g. cloud base height and cloud tops, temperature and so on. From this information one could at least estimate the adiabatic LWC, which give us a range for the liquid water mixing ratio and, therefore, the maximum error for the calculated buoyancy.

We agree on that and have included this information in the revised manuscript. We understand that this additional information has an important value for the reader. We have added the information about the sampled cloud fields to Table 2 in the manuscript which lists the measurement flights of the campaign. The influence on the buoyancy is now discussed in Sec. 2.4.2.

P4, 112: Please discuss a little bit more in detail why you didn't simply applied criteria used for previous observations. This would have the advantage to better compare the results with each other. You should have good reasons to introduce new criteria!

We have addressed this point with a close look at the previous observations. This is indeed important for the cloud definition. However, the definition of the subsiding shell is still limited to the median distribution, while we refrain from investigating the individual transects - as suggested by the reviews. The respective discussions in the new manuscript are at p. 4 I 4-17 f or the cloud definition and at p. 4 I 22-33 for the subsiding shell)

Sec 2.4.2 "LWC – in particular for non-adiabatic regions such a cloud boundaries is a highly fluctuating parameter. To assume a constant value is a very strong simplification. Please discuss in detail the consequences and a maximum error for the derived buoyancy. Without such a discussion the presented buoyancy is highly questionable. I suggest ignoring the LWC and discussing the maximum error for B; this might be more straightforward compared to use a constant value for LWC – the error will be small and not alter your results. The variation of LWC as seen in Wang et al describe more the deviation from the adiabatic LWC which itself is a function of height and cloud base temperature."

The influence on the buoyancy is now discussed in the respective section (p6 I 9-16). We calculate the adiabatic LWC gradient from the measured profile data during the flights and as suggested give an estimation of the maximum errors. We have also followed the suggestion to show the buoyancy calculation without the LWC influence in Fig. 10 c).

Discussion, line 23ff: "You mentioned that a certain fraction of the sampled clouds do not show a subsiding shell on both sides. Based on this finding I have serious concerns how representative an estimated mass flux distribution is? One should remember that a flight through a cloud is one single realization and more a "spaghetti-like" penetration."

Indeed, the calculation of the vertical mass flux along the flight path leads to a difference when compared to the aerial calculation, which are possible with LES models (Heus et al. 2009). However, it is a common method also described in earlier publications (e.g., Yang et al., 2016). A single realization of the mass flux as determined in our paper is certainly not representative for 'a cloud', but averaging (over many spaghetties) should at least provide an estimate of the potential magnitude and direction. This is also important for the cloud properties in Figs. 7 and 10 where we only show the statistical distributions (i.e., the medians and percentiles). On the reviewer's suggestion we have emphasized this more (and hopefully better) in the revised manuscript.

Our conclusion of the calculation describes a significant contribution of the compensating vertical mass flux in the near surrounding of the cloud. This is in concordance with the LES results even though we find it to be the result of the elevated frequency of downdrafts in the vicinity of the clouds.

P11, line 1 "We find a positive correlation of vertical wind and buoyancy (i.e., r \_ 40%). Near the cloud gaps this indicates mixing of cloud air with environmental air" I cannot follow this logic; the first part of this statement simply states that about 40% of the data shows an actively growing cloud (following Katzwinkel's nomenclature) but how can you conclude that this means mixing around cloud gaps? Cloud gaps are most probably the consequence of mixing. Please explain your conclusion. The intention of next statement is also not clear; it is the nature of turbulence that up- and downdrafts can be observed close to each other so why should one be surprised to find this in cumulus clouds? Maybe this is simply a misunderstanding on my side"

We have modified the argumentation in the revised manuscript (p 12 I. 34ff).

P11, 112" On the downwind side of active clouds the broad region of downdrafts is explainable by a humidity halo as observed by Perry and Hobbs (1996). Why can broad regions with downdrafts be explained by humidity halos. By the way, if you cannot measure

LWC but instead define your cloud boundary by relative humidity you cannot identify humidity halos, which are characterized by almost saturated conditions but no cloud droplets."

We agree, that this needs further explanation. As discussed above, there is a small chance that we assume a cloud where already the region of a humidity halo has started, but not the opposite way. Therefore, when we measure enhanced humidity compared to the environment we can detect this region. The humidity halo is seen as a mixture of cloud and environmental air. Therefore, it can have significant subsaturation and enhanced humidity. The evaporation of the cloud droplets leads to cooling and negative buoyancy (p9 I 15-19, p12 I 27-34). The correlations of buoyancy and vertical wind in the analysis of the downdrafts have been calculated and added (Table 7) to support these conclusions.

*"It was quite often concluded that your observations are similar to previous observations* – so the reader is left with the question "What is new in your study?' *An investigation of size, turbulence statistics and scaling with the cloud size in future research is desirable to understand the dynamics of shallow convection over land.' Why not starting with answering these questions in this paper?"*

We have adopted this suggestion and expanded the revised manuscript by the analysis of the up- and downdrafts in the vicinity of the clouds (Sec. 3.4). We hope that this has added a valuable additional and new apsect to the revised paper.

[revised manuscript text omitted]